# The influence of lateral flow on land surface fluxes in southeast Australia varies with model resolution

Anjana Devanand[1,2,3], Jason P. Evans[1,2], Andy J. Pitman[1,2], Sujan Pal[4], David Gochis[5,6], Kevin Sampson[5]

[1]Australian Research Council Centre of Excellence for Climate Extremes, UNSW Sydney, NSW 2052, Australia
[2]Climate Change Research Centre, UNSW Sydney, NSW 2052, Australia
[3]CSIRO Environment, Canberra, ACT 2601, Australia
[4]Environmental Science Division, Argonne National Laboratory, Lemont, Illinois, 60439, USA
[5]National Center for Atmospheric Research, Boulder, Colorado, 80301, USA
[6]Airborne Snow Observatories, Inc., Mammoth Lakes, California, 93546, USA

*Correspondence to*: Anjana Devanand (anjana.devanand@csiro.au)

**Abstract.** Land surface models (LSMs) used in climate models typically represent surface hydrology as one-dimensional vertical fluxes, neglecting the lateral movement of water within and between grids. It is assumed that lateral flow of water has a negligible impact on land surface states at climate modelling resolutions of a few tens of kilometres. However, with increases in model resolution, it may be necessary to include lateral flow in LSMs as satellite observations indicate the influence of this process on ecohydrological states, particularly in water limited regions. Lateral flow has not been modelled in Australia, but there is some evidence that this process exerts a dominant influence on vegetation variability in arid and semi-arid Australia. Here we use standalone WRF-Hydro simulations to quantify the influence of overland and shallow subsurface lateral flow on surface fluxes in southeast Australia, and the impact of model resolution on the results. We perform LSM simulations at 1-km, 4-km, and 10-km resolutions, with and without lateral flow, to assess the changes in evapotranspiration. Our results show that lateral flow increases evapotranspiration near major river channels in LSM simulations at 4- and 1-km resolutions, consistent with high-resolution observations. The largest changes occur in the warm season after a wet winter, with magnitudes of 50% or more in some areas. However, the 1-km resolution simulations also exhibit a widespread pattern of drier ridges, different from the coarser resolutions. At 10-km resolution the increases in evapotranspiration are confined to the mountainous regions. Our results suggest that it may be necessary to include lateral flow in LSMs for improved simulations of droughts and future water availability at resolutions higher than 10 km.

## 1 Introduction

Land surface models (LSMs) were developed to represent surface thermodynamic, hydrological and biogeochemical processes and provide lower boundary fluxes to the atmosphere in coupled climate model simulations. Land hydrology in LSMs is conceptualised as one-dimensional vertical fluxes that partition precipitation reaching the ground into evapotranspiration, runoff, and storage change (Clark et al., 2015). While some aspects of spatial land heterogeneity related to variation in land cover, soils and vegetation types are incorporated in this one-dimensional framework (Lawrence et al., 2019; Walters et al.,

2019), the lateral flow of water within and between model grid cells is typically not represented (Clark et al., 2015; Fan et al., 2019). In some cases, lateral flow is modelled using routing models uncoupled from LSMs as a subsequent part of the modelling system, to obtain streamflow (Li et al., 2013; Yamazaki et al., 2014). The eco-hydrological effects of lateral flow manifests at spatial scales of tens of meters to kilometres (hillslope scales) and this process is assumed to have negligible influence on states at LSM grid scales (typically a few tens to around 100 km) (Fan et al., 2019). Currently, climate modelling resolutions are increasing with advancements in computing (Demory et al., 2020; Lucas-Picher et al., 2021) and LSMs have expanded to represent a range of process linked to vegetation dynamics (Fisher et al., 2018), nutrient cycling (Sun et al., 2021), fire dynamics (Curasi et al., 2024) and aspects of water management including irrigation (Evans and Zaitchik, 2008; Pokhrel et al., 2016). Many of these processes are linked to hydrologic states in the LSM (Blyth et al., 2021). As summarised by Fan et al. (2019), processes such as topography driven drainage and variations in solar insolation are understood to be fundamental organisers of eco-hydrological states at finer spatial resolutions and need to be incorporated into LSMs when applied at fine spatial resolutions.

While lateral flow is not directly observed, other lines of evidence indicate the relevance of this process. Modelling studies using LSMs without lateral flow attribute reduced soil storage, faster subsurface drainage, and faster shutdown of evapotranspiration and streamflow to the lack of lateral flow (Fan et al., 2019). Satellite observations show vegetation patterns consistent with lateral water flow (Chen et al., 2016; Tai et al., 2020), and solar insolation differences (Pelletier et al., 2018). The water, energy and carbon exchanges between the land and atmosphere are strongly coupled to vegetation activity at regional and global scales (Duveiller et al., 2018; Forzieri et al., 2020). Norton et al. (2022) report that areas with higher hydrologic connectivity account for a larger proportion of the land carbon uptake in arid and semi-arid Australia, and lateral flow drives the variability in vegetation productivity in these regions. Thus, there is evidence that lateral flow processes in LSMs may be important in modelling water, energy and carbon cycles.

There have been several model developments to include lateral flow processes in LSMs, and they fall into two broad categories (1) representation of lateral subsurface flow, (2) representation of lateral overland flow in addition to subsurface flow. Models that represent lateral subsurface flow range in complexity and include (1a) coupling a 3-D groundwater model to an LSM (Bisht et al., 2018; Zhu et al., 2024), (1b) quasi-3D approaches that incorporate a source term to account for lateral flow in the vertical solution of soil water fluxes (Felfelani et al., 2021; Qiu et al., 2024), and (1c) more simplistic sub-grid representative hillslopes with specified connectivity that move water laterally within a model grid, but not between grids (Hazenberg et al., 2016; Swenson et al., 2019; Zhang et al., 2024). The three main modelling systems that represent lateral overland flow in addition to subsurface flow are (2a) HydroBlocks (Chaney et al., 2021), (2b) WRF-Hydro (Gochis et al., 2020), and (2c) ParFlow-Common Land Model (ParFlow-CLM) (Naz et al., 2023). HydroBlocks and WRF-Hydro represent the lateral movement of overland flow from infiltration and saturation excesses at fine spatial resolutions (of order 100 m). The HydroBlocks model has been developed over the past few years and has not been widely applied. WRF-Hydro has been developed over the past two decades for streamflow forecasting (Cosgrove et al., 2024; Gochis and Chen, 2003), but a few studies have applied it to study the influence of lateral flow (Arnault et al., 2021; Lahmers et al., 2020; Yang et al., 2021).

ParFlow-CLM simulates deeper groundwater flow (up to about 100 m below ground) as well as overland flow using a single framework at spatial resolutions of about 1 to 10 km (Keune et al., 2016; O'Neill et al., 2021). This model requires more subsurface information than the other two modelling systems and is typically used without calibration due to higher computational requirements. A few studies have used ParFlow-CLM to examine the influence of lateral flow (Keune et al., 2016; Maxwell and Condon, 2016). It is worth noting that this summary pertains to modelling systems that includes lateral flow processes in LSMs used for Earth system modelling. Other models of lateral flow have been developed in the domain of integrated surface subsurface hydrological modelling to understand watershed system function (Bhanja et al., 2023; Brunner and Simmons, 2012). But these formulations have not been used in LSMs, likely due to the challenges in coupling them with LSMs.

Several studies have used LSMs with lateral flow to quantify the impact on land surface and climate states. Swenson et al. (2019) used global LSM simulations at coarse spatial resolutions with subsurface intra-grid lateral flow and found that lateral flow causes differences in evapotranspiration between upland and lowland hillslope columns, and that the largest differences occur in arid and semi-arid regions. Subsequently, regional models were used to understand the influence of lateral flow on land surface states at higher spatial resolutions in parts of United States, Europe, Africa and China. These studies report that while increases in regional mean soil moisture and evapotranspiration (ET) induced by lateral flow are small, the changes are spatially heterogenous (Chaney et al., 2021; Fersch et al., 2020; Lahmers et al., 2020; Qiu et al., 2024; Zhang et al., 2024). In urban areas, fine scale heterogeneity of impervious areas and open spaces may induce substantial changes in the surface energy balance (Alexander et al., 2024; Reyes et al., 2016). On regional scales, including lateral ground water flow is reported to increase the proportion of ET from transpiration over the United States (Maxwell and Condon, 2016). Coupled regional land - atmosphere simulations indicate that lateral flow can influence convective organisation (Lahmers et al., 2020) and recycled precipitation (Arnault et al., 2021; Zhang et al., 2021) in some regions. In Australia, the influence of lateral flow on land surface states has not been examined in regional LSMs.

Overall, the evidence suggests that gradient driven lateral flow convergence can modulate regional water, energy and carbon fluxes and that it may be necessary to model this process in high-resolution LSMs, particularly in water limited regions. While the influence of hydrologic connectivity is visible in vegetation patterns in semi-arid Australia, the impact of lateral flow in this region has not been quantified. This study aims to understand the influence of lateral flow on land surface states in southeast Australia, and the impact of model resolution on the results. We use standalone LSM simulations at varying spatial resolutions to examine the changes in ET and surface water partitioning due to lateral flow. In standard LSM simulations, surface and subsurface runoff are removed from the system at the end of each model time step. In LSM simulations with lateral flow representation, runoff remains in the system and continues to affect other water cycle components such as ET and soil moisture.

## 2 Methods

### 2.1 Experiment Design

We perform standalone simulations using WRF-Hydro version 5.2 (Gochis et al., 2020) with the Noah-MP LSM to study the influence of lateral water flow on surface fluxes in a domain that covers an area in southeast Australia (Fig. 1). There is
substantial topographic variation in the domain with low lying areas in the northwest and mountainous regions with elevations above 1000 m in the southeast. The domain includes upper reaches of the Murray River system that flows west across the region (Appendix Fig. A1a). Figure A1(a) shows the surface water catchments and major rivers in the region – the Murrumbidgee, Upper Murray, Ovens and Goulburn rivers. In the results Sect. 3.2.3, we analyse the modelling results in more detail in the connected basins outlined in blue in Fig. 1 - Upper basins (the combined Upper Murray and Kiewa basins), Ovens,
and Murray Riverina.

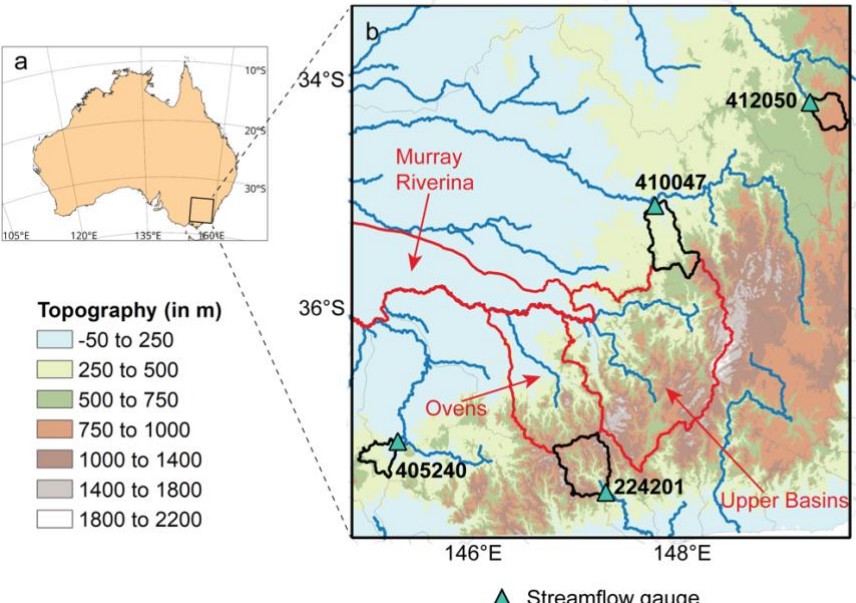

Figure 1: (a) The WRF-Hydro model domain in southeast Australia. (b) Features in the domain. Background shading indicates
topography, and the surface water catchments (black outlines) that drain into the streamflow gauges (blue triangles) used for calibration are marked on the map. The basins outlined in red (Upper Basins, Ovens, and Murray Riverina) are used to analyse the influence of lateral flow in basins with varying topographic characteristics. The network of major rivers (blue lines) based on data from Geoscience Australia (https://pid.geoscience.gov.au/dataset/ga/42343) are shown in panel b.

The WRF-Hydro modelling system consists of a column LSM (here we use Noah-MP) that simulates the vertical exchange of
water and energy at the surface at resolutions of 1 to 10 km, coupled to routing modules that simulate lateral flow at finer resolutions (100s of meters). WRF-Hydro has the capability to simulate overland, shallow subsurface, and channel flows on

the fine resolution routing grid. The subsurface lateral flux in the saturated portion of the soil column is calculated based on hydraulic gradients using the method documented in Wigmosta et al. (1994) and Wigmosta and Lettenmaier (1999), implemented in the Distributed Hydrology and Soil Vegetation Model (Gochis et al., 2020). Overland and channel flow is
calculated using the diffusive wave formulation, using Manning's equation as the resistance formulation (Gochis et al., 2020). In the simulations presented in this paper, overland and shallow sub-surface flows are modelled, and these processes feedback to influence soil moisture and surface fluxes in the LSM. The water that flows into the channel grids is routed downstream to model streamflow and does not feedback to affect the LSM soil moisture. In other words, channel leakages are not represented in the model. The WRF-Hydro modelling system also has the functionality to specify a conceptual representation of baseflow
by passing the underground runoff from the LSM directly into the channel network, but this is not used in the simulations presented in this paper. We calibrate model parameters to match streamflow at selected gauges. Based on preliminary calibration results, the baseflow representation is turned off in our simulations. It is worth noting that lateral saturated sub-surface flow is being modelled in our simulations as described above, despite baseflow being turned off.

Here we use the standard Noah-MP LSM which has a constant soil depth of 2 m with vertically homogeneous soil parameters.
This formulation can contribute to biases in runoff and evapotranspiration, which may be ameliorated by incorporating variable and higher soil depths, groundwater processes, and vertical soil heterogeneity (Gochis et al., 2010; Barlage et al., 2015; Wu et al., 2021; Yimam et al., 2025) in the modelling framework. These aspects have not been explored in this study as they are outside the scope of the work presented here.

Our experiment set consists of simulations with and without lateral flow at different LSM and routing resolutions (Table 1).
The higher order channels in the routing grid used for modelling are identified from digital elevation data and they match the river network in the region available from other sources (Appendix Fig. A1). The Noah-MP parameterisations used are the defaults in WRF-Hydro (Gochis et al., 2020) and are listed in Appendix Table A1. We estimate the differences between the 'LAT' and 'CTL' simulations at the same resolutions to quantify the influence of lateral flow and use the suite of experiments to understand the variation with model resolution. We simulate years 2013 to 2017 and analyse the changes in ET during 2015-
12 to 2017-11 broken into individual seasons, discarding the first ~3 years as spin-up.

**Table 1: List of simulations in the experiment set up**

| Simulation type | Simulation name | Land surface model resolution (km) | Routing | Routing resolution (m) |
|---|---|---|---|---|
| CTL (Noah-MP) | CTL1 | 1 | Off | - |
| | CTL4 | 4 | | - |
| | CTL10 | 10 | | - |
| LAT | LAT1-250 | 1 | On | 250 |
| | LAT1-100 | 1 | | 100 |

| (Noah-MP + WRF-Hydro) | LAT4-250 | 4 | | 250 |
|---|---|---|---|---|
| | LAT4-100 | 4 | | 100 |
| | LAT10-250 | 10 | | 250 |
| | LAT10-100 | 10 | | 100 |

The analyses years 2016 and 2017 consist of a wet year followed by a dry year in this region. From the Australian Gridded Climate Data (AGCD) (Jones et al., 2009), the domain average annual rainfall in 2016 is 940 mm, which is in the top decile of the long term record since 1911. In contrast, the domain average annual rainfall in 2017 is 640 mm which is about the $30^{th}$ percentile. The domain average annual temperatures during both years are 15 degrees Celsius, which is slightly more than 1 degree above the 1911-1960 mean.

## 2.2 Datasets

### 2.2.1 Geographic data

Static datasets from different sources are used to provide land information to the model. We use the global Hydrosheds Digital Elevation Model (DEM) (Lehner et al., 2008) available at a resolution of 90 m (https://www.hydrosheds.org/) to delineate the flow paths and accumulation areas required for the routing modules. The landcover of year 2014-15 from the Dynamic Land Cover Dataset version 2.1 (DLCDv2.1) available from Geoscience Australia (Lymburner et al., 2011) is used to provide land cover information for the LSM, and the soil data from the Food and Agriculture Organisation, which is available as part of default geographic datasets for WRF are used to provide soil type information. Preliminary experiments indicated that the model simulations of streamflow are particularly sensitive to soil representation. We analysed simulations using alternate soil types specified based on soil attributes data available from the Terrestrial Ecosystem Research Network (TERN) (Searle, 2021), but finally used the default soil data as it resulted in a closer match to observed streamflow. The better streamflow simulations obtained using the default soil dataset rather than the TERN dataset may be surprising, as the TERN data, which utilises regional observations (Teng et al., 2018), likely provides more accurate soil information over Australia. This is possibly because the modelled influence of soils on surface water partitioning in each land column relies on soil parameters in addition to soil type. The land model uses parameters (such as moisture at saturation, field capacity, wilting point, saturated hydraulic conductivity) for each soil type, the default values for which are defined based on scarcely available field observations in various regions (Kishné et al., 2017). Our results suggest that regional measurements that can be used to refine the default parameters values in conjunction with the regional soil datasets, such as the TERN dataset, may be necessary to obtain improved simulations.

**2.2.2 Meteorological forcing data**

Standalone simulations of WRF-Hydro need meteorological forcing data consisting of precipitation, 2-m air temperature,
humidity, near surface winds, surface pressure, downwelling shortwave and longwave radiation variables at least at 3-hourly
temporal resolution (Gochis et al., 2020). Some of the required forcing variables over Australia are available from an
observation based gridded dataset, the Australian Gridded Climate Data (AGCD) (Jones et al., 2009). However, these
observations are available only at daily timescales, which are insufficient. Therefore, in this study, we force the simulations
with hourly data from the ERA5-Land reanalysis dataset (Muñoz-Sabater et al., 2021). We correct the precipitation and
temperature from the reanalysis forcing to match monthly AGCD data to obtain more realistic climate forcings, which is
necessary as we calibrate the model to simulate observed streamflow. The monthly accumulated precipitation and mean
temperature from ERA5-Land reanalysis are scaled to match the AGCD observations for the corresponding month at each
grid. This correction is performed as AGCD is a high-quality historical data developed by applying topography resolving
analyses methods to in situ observations (Jones et al., 2009) and is expected to be more accurate than reanalysis data. The
corrected forcing data would thus match the AGCD closely at monthly timescales, while using the sub daily pattern of variation
in the ERA5-Land reanalysis. Compared to ERA5-Land, the AGCD precipitation exhibits higher spatial variation primarily
over the mountainous areas in the domain (Fig. A2), and these differences can have substantial effect on infiltration rates in
land simulations (Sampson et al., 2020). Application of the monthly scaling correction induces localised increases and
decreases in hourly precipitation rates from ERA5-Land as shown in Fig. A2(c).

**2.2.3 Calibration and evaluation data**

WRF-hydro model parameters are calibrated to match 3-day mean streamflow at four Hydrologic Reference Stations (HRS)
in the domain (gauges 410047, 412050, 224201, 405240). Observed streamflow at the HRS locations are available from the
Australian Bureau of Meteorology (Zhang et al., 2013). The catchments and gauges used in calibration are shown in Fig. 1.
We validate the modelled ET from the WRF-Hydro simulations against two observation-based ET datasets. We use actual ET
from a satellite based high-resolution (30 m) monthly dataset estimated using the CMRSET (CSIRO MODIS Reflectance-
based Scaling EvapoTranspiration) algorithm version 2.2 (Guerschman et al., 2022; McVicar et al., 2022), referred to as the
CMRSET data hereafter. We also use a merged observationally constrained ET product with a relatively coarser spatial
resolution of 0.25 degree, the Derived Optimal Linear Combination ET (DOLCE) version 3 (Hobeichi et al., 2022), hereafter
referred to as the DOLCE dataset. The DOLCE dataset also provides uncertainty estimates.

**2.3 Calibration of model parameters**

We calibrate sixteen WRF-Hydro model parameters as listed in Table 2. Eleven of the parameters are known to influence
streamflow, based on prior applications of WRF-Hydro in the United States (Pal et al., 2023; Wang et al., 2019). Additionally,
we calibrate the model parameter LKSATFAC, which also influences modelled streamflow in our simulations based on

preliminary assessments. LKSATFAC is a multiplier on the lateral hydraulic conductivity in the model, and we calibrate this parameter by soil type. We use the Parameter ESTimation (PEST) tool (Doherty, 2016) for calibration. We apply the parallel PEST tool using an optimisation function to reduce the biases in 3-day mean streamflow during a 45-day period from 2016-09-01 to 2016-10-15 (i.e., 15 observational data points) at all four streamflow gauges shown in Fig. 1. Calibration of WRF-Hydro is computationally intensive and involves choices that may be aligned to the purpose of the simulations. Here we study the influence of lateral flow on seasonal timescales and hence the main purpose of calibration is to obtain better streamflow outcomes on monthly to seasonal timescales, rather that improved simulations of daily scale streamflow events. The streamflow in the domain primarily occurs in the cool season (May to October), and model simulations using default parameter values exhibit biases during these high flow months (Fig. 2). However, preliminary results showed that event-based calibration to high flow days did not translate to improved monthly flows indicating that it is necessary to use a period at least of the order of a month, which includes both high and low flow days at the four gauges for calibration. As a 45-day period is computationally feasible, and yields reasonable outcomes at monthly to seasonal timescales, this length of time is chosen for calibration. The daily streamflow data is smoothed by aggregating to 3-day flows to dampen the effect of individual high flow days. Eight PEST iterations are performed to identify the optimum parameter values. The calibration is performed for a simulation with an LSM resolution of 4-km and routing resolution of 250 m (LAT4-250). We use the same calibrated parameter values for the simulations at the other model resolutions listed in Table 1. To quantify the improvements in biases and Nash Sutcliffe Efficiency (NSE) resulting from the calibration, we compare modelled streamflow using the calibrated and default parameter values in longer simulations spanning two years, 2016-17.

**Table 2: The model parameters included in the calibration, their default values, lower and upper bounds, and the optimum values identified through calibration.**

| Parameter name | Parameter description | Default | Lower bound | Upper bound | Calibrated optimum value |
|---|---|---|---|---|---|
| xslope1 | Determines the bottom drainage from the LSM soil column | 0.1 | 1e-4 | 1 | 0.0128 |
| REFDK | Reference for soil conductivity | 2e-6 | 1e-8 | 1e-5 | 1.54e-8 |
| REFKDT | Soil infiltration parameter | 3 | 0.01 | 5 | 1.82 |
| ovn1 | Overland roughness of land use class 'evergreen broadleaf forest' | 0.2 | 0.1 | 0.3 | 0.287 |
| ovn2 | Overland roughness of land use class 'dryland cropland and pasture' | 0.035 | 0.015 | 0.06 | 0.015 |

| | | | | | |
|---|---|---|---|---|---|
| LKSATFAC3 | Multiplier on lateral hydraulic conductivity of soil class 3 'sandy loam' | 1000 | 10 | 10000 | 8617 |
| LKSATFAC6 | Multiplier on lateral hydraulic conductivity of soil class 6 'loam' | 1000 | 10 | 10000 | 2312 |
| LKSATFAC7 | Multiplier on lateral hydraulic conductivity of soil class 7 'sandy clay loam' | 1000 | 10 | 10000 | 173 |
| LKSATFAC9 | Multiplier on lateral hydraulic conductivity of soil class 9 'clay loam' | 1000 | 10 | 10000 | 37 |
| LKSATFAC12 | Multiplier on lateral hydraulic conductivity of soil class 12 'clay' | 1000 | 10 | 10000 | 10000 |
| mann1 | Manning's n of stream order 1 | 0.55 | 0.35 | 0.6 | 0.35 |
| mann2 | Manning's n of stream order 2 | 0.35 | 0.15 | 0.35 | 0.15 |
| mann3 | Manning's n of stream order 3 | 0.15 | 0.08 | 0.15 | 0.127 |
| mann4 | Manning's n of stream order 4 | 0.10 | 0.05 | 0.15 | 0.0973 |
| mann5 | Manning's n of stream order 5 | 0.07 | 0.02 | 0.10 | 0.1 |
| mann6 | Manning's n of stream order 6 | 0.05 | 0.015 | 0.10 | 0.05 |

## 2.4. Water balance in the simulations

The simulated water cycle components are used to understand the influence of lateral flow on surface water partitioning. In control simulations using Noah-MP LSM without lateral flow, incoming precipitation is partitioned into ET, surface runoff (variable name: *sfcrnoff*), underground runoff (variable name: *udgrnoff*), and changes in soil moisture in the four layers (0-10 cm, 10-40 cm, 40 – 100 cm, 100 – 200 cm) of the soil column. The volumetric soil moisture in each layer converted to water depths are used to estimate the total soil moisture change for water balance calculations. The total runoff is estimated in two ways (a) as the sum of the surface and underground runoff components, and (b) as the residual of precipitation after ET and soil moisture changes. We use these components to demonstrate the closure of the water balance in the control simulations. In simulations including lateral flow, the total runoff consists of channel flow, overland flow and underground runoff components simulated on the fine resolution routing grid. The runoff terms on the routing grid are not written to output files to reduce computational expense. Hence, we estimate the total runoff in the lateral flow runs as the residual of precipitation after ET and soil moisture changes closing the water balance in a manner consistent with the control simulations.

## 3 Results

### 3.1 Evaluation against observations

#### 3.1.1 Streamflow

Figure 2 compares the observed cumulative monthly streamflow during 2016-17 with modelled streamflow in simulations using default and calibrated parameter values in LAT4-250 simulations. The use of calibrated parameters reduces biases and improves NSE at all four gauges. In simulations using calibrated parameters, the NSE are higher than 0.65. The streamflow at gauges 224201 and 410047 exhibit biases of around 15%, whereas the streamflow at gauges 412050 and 405240 are underestimated by 52-57%, even with the calibrated parameters. This underestimation is likely related to the specification of

baseflow in the simulations. The conceptual baseflow representation is turned off in our simulations to achieve a better representation of streamflow, considering all four gauges. We found that including baseflow improves the streamflow at gauges 412050 and 405240, while deteriorating the others. It may be possible to improve modelled streamflow through the implementation of baseflow representations that vary between catchments, which is outside the scope of this study. As mentioned in the 'Methods' section, we use default soil data available from the geographic dataset of WRF in our simulations

based on analyses of streamflow using alternate soils. The sensitivity of streamflow to soil data is shown in Appendix Fig. A3. The sensitivity to soil specifications and baseflow noted here indicate the potential for spatially variable parameters to improve simulations in this region, which may be explored in future work.

Appendix Figure A4 shows the monthly streamflow during 2016-17 in simulations at different resolutions with all simulations using the calibrated parameters from the LAT4-250 run. The streamflow magnitudes in 1-km and 4-km LSM resolutions are

similar to the streamflow from the calibrated LAT4-250 run. However, the 10-km resolution LSM run exhibits large biases at three out of the four gauges, probably due to the coarse grid resolution. Recalibrating model parameters at 10-km resolution may reduce the biases at this resolution. However, in this study, we use the calibrated parameters from the LAT4-250 simulation for all simulations in the experiment set, so that the differing model parameters do not affect our inferences about the impact of lateral flow.


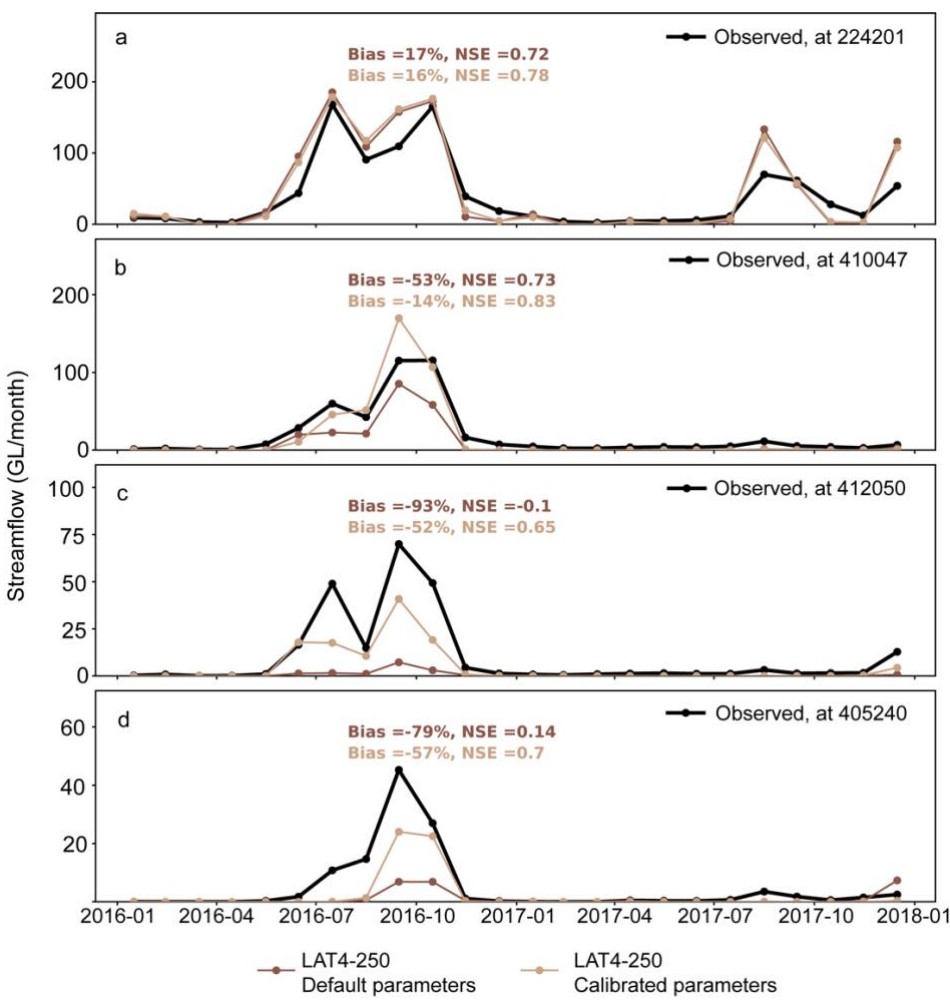

**Figure 2: Improvements in streamflow with calibration.** Simulated cumulative monthly streamflow using default and calibrated parameters (coloured lines) compared to observed streamflow (black lines) at the gauge locations (in GL/month) in simulations with an LSM resolution 4 km and routing resolution 250 m (LAT4-250). The biases and NSE of the simulations are indicated by the numbers in the corresponding colours in each panel.

### 3.1.2 Evapotranspiration

We compare the simulated domain average monthly ET (domain shown in Fig. 1b) during 2016-17 with estimates from the CMRSET and DOLCE datasets in Fig. 3. The DOLCE dataset includes uncertainty estimates that provide a range for the ET in the domain. The simulated timeseries of ET are within the range from the DOLCE product 75% of the time, and slightly outside this range the rest of the time (6 out of 24 months). When compared to the satellite-based CMRSET data, the simulations exhibit some biases. During the summer months, there are positive biases of about +30% in the simulations with respect to the CMRSET data. The simulated ET is also at the higher end of the range indicated by the DOLCE product in summer. In winter and early spring of year 2017 (June to October of 2017), the simulations exhibit negative deviations of about -20% from the CMRSET estimate and are also near the lower end of the range from the DOLCE dataset. The negative

biases in ET in dry year 2017 in our simulations are consistent with previously reported systematic biases in land surface models (Ukkola et al., 2016).

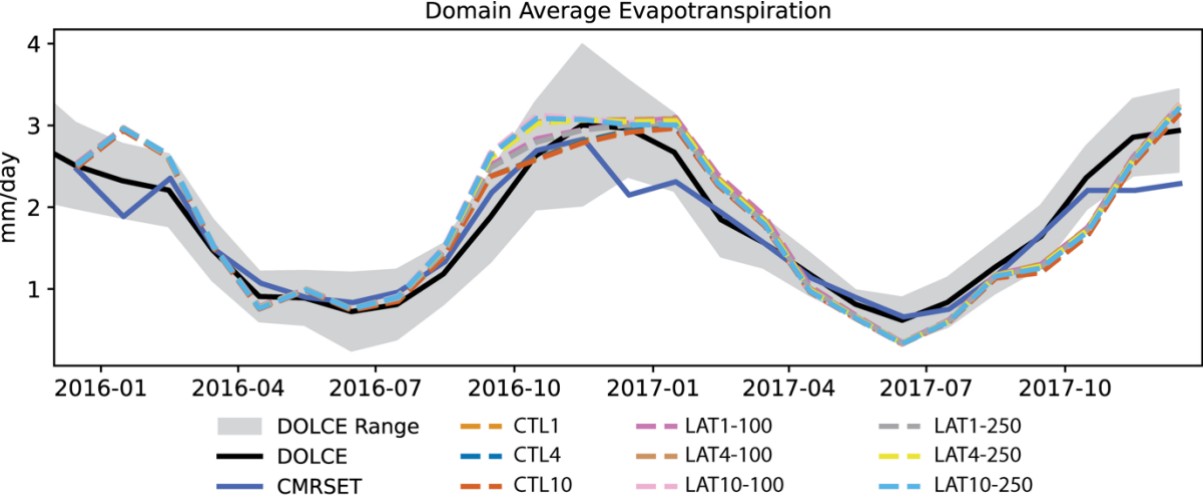

**Figure 3: Comparison of the domain average ET (in mm/day) during 2016-17 in the model simulations with CMRSET and DOLCE datasets. During some months, the different dashed lines are not differentiable because the domain averages from the different**
**simulations overlap.**

### 3.2 Influence of lateral flow

### 3.2.1 Spatial changes in evapotranspiration

While the domain average ET from simulations at different resolutions are similar (Fig. 3), there are differences in the spatial patterns. Figure 4 shows the spatial pattern of the changes in seasonal mean ET with inclusion of lateral flow in simulations at
varying LSM resolutions, using a 250 m routing grid. The ET changes in simulations using a 100 m routing grid are shown in Fig. 5. Including lateral flow results in largest changes in spring (SON) and summer (DJF). In spring, the domain average ET increases by about 0.1 to 0.2 mm/day (4 to 9%). The domain average changes in summer are slightly smaller in magnitude. The spatial pattern of changes in ET varies with the resolution of the LSM. At a resolution of 1-km (Fig. 4h), there is a reduction in ET in upstream areas that do not exist in the coarser resolution runs (Fig. 4l, 4p). In summer, the reduction in ET in upstream
ridges are visible at all resolutions, but more widespread and prominent in the 1-km LSM simulation (Fig. 4e). At LSM resolutions of 1-km and 4-km, the patterns of major rivers in the domain, the Murrumbidgee, Ovens, and Goulburn rivers (river network is shown in Appendix Fig. A1) are visible in the changes in ET. At 10-km, the influence of drainage networks is not visible in the ET changes. The Murray Riverina region located downstream of the Ovens and Goulburn rivers experience the largest changes in ET in the 4-km and 1-km runs. In winter and autumn, the ET changes are smaller than in the other two
seasons with minor increases near the river channels.

The resolution of the routing grid has a smaller effect on these results, as the general pattern of ET changes in simulations using 250 m and 100 m routing grids at the same LSM resolution are similar (Fig. 4 and 5). In terms of domain averages, simulations with 100 m routing result in slightly stronger changes compared to the 250 m routing grid, at the same LSM resolutions. The differences in spatial patterns also indicate that larger changes in ET tend to be located further upstream in simulations using a 100 m routing grid.

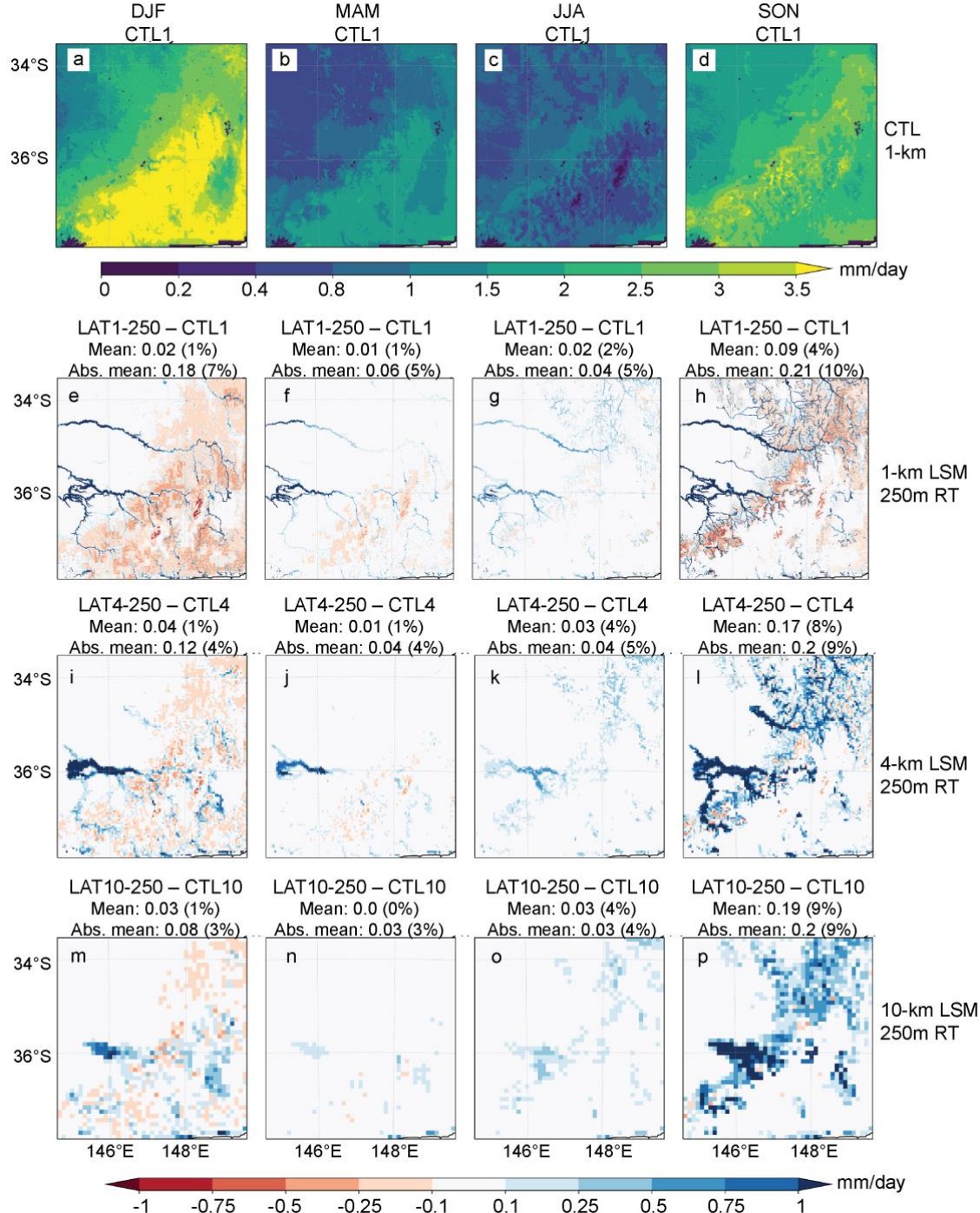

**Figure 4: Spatial patterns of seasonal mean ET at various LSM resolutions with 250 m routing (a-d) Mean seasonal ET in the CTL1 simulation. The mean seasonal changes in ET with inclusion of lateral flow (in mm/day) (e-h) LAT1-250 minus CTL1 (i-l) LAT4-250 minus CTL4, (m-p) LAT10-250 minus CTL10. The mean and absolute mean changes averaged over all the grids in the domain in mm/day and percentage are indicated in the panel titles of (e-p). The mean seasonal patterns of ET in CTL4, and CTL10 simulations are similar to that shown in (a-d) and are thus not included in this Figure.**


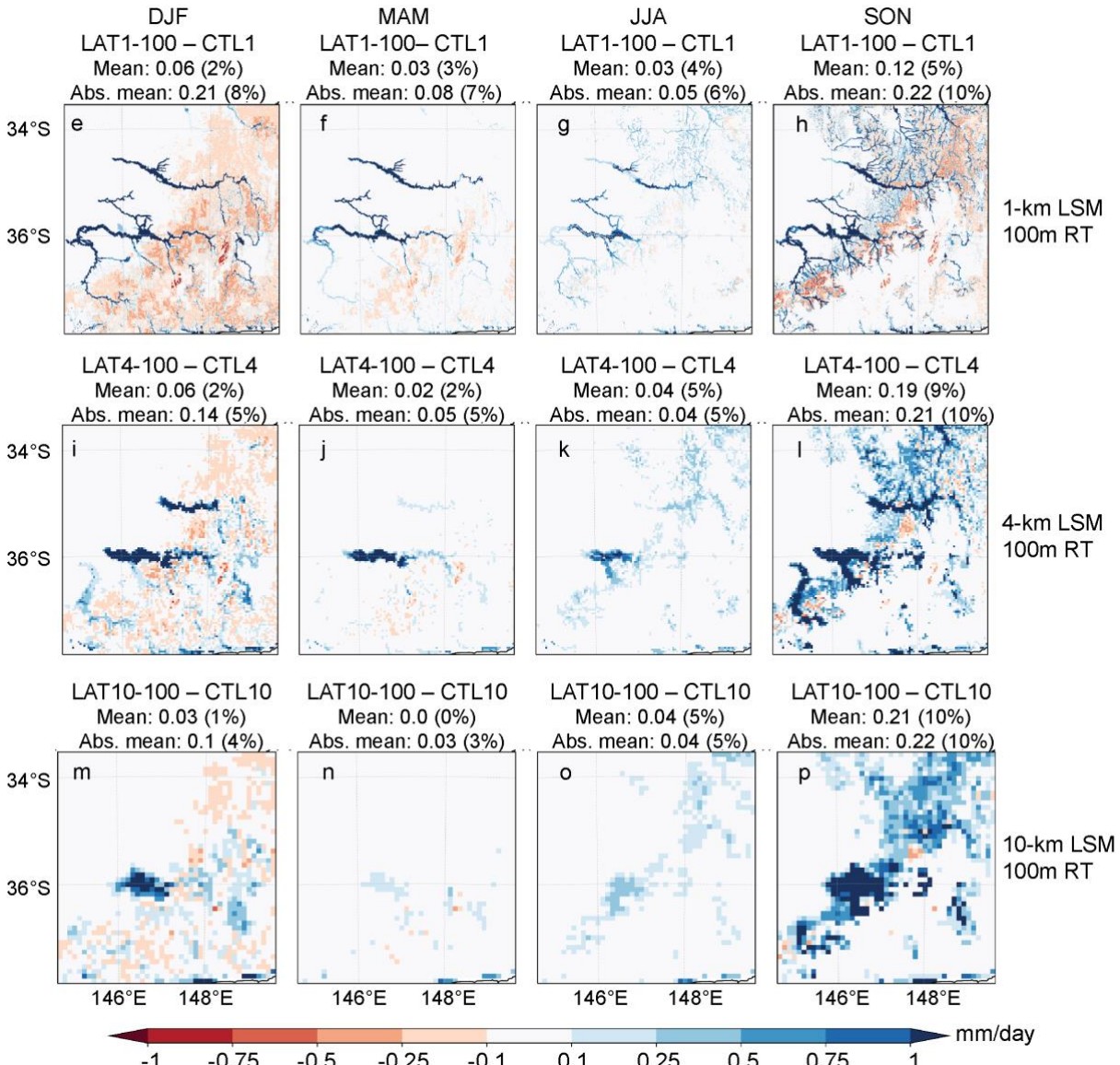

**Figure 5: Similar to Figure 4(e-p) but for simulations with 100 m routing. The mean seasonal changes in ET with inclusion of lateral flow (in mm/day) (a-d) LAT1-100 minus CTL1 (e-h) LAT4-100 minus CTL4 (i-l) LAT10-100 minus CTL10. The mean and absolute mean changes averaged over all the grids in the domain in mm/day and percentage are indicated in the panel title.**

Appendix Table A2 lists the domain average precipitation, ET, runoff components, and the change in soil moisture from the nine simulations. In the control runs, total runoff estimated as (a) the sum of the surface runoff and underground runoff components matches (b) the residual of precipitation after ET and total soil moisture change, demonstrating the closure of water balance. In simulations including lateral flow, total runoff is estimated as the residual of precipitation after ET and total soil moisture change. In terms of domain average water balance, the redistribution of water through the inclusion of lateral

flow facilitates increases in ET and soil storage in the domain, and a decrease in total runoff. In control simulations without lateral flow (CTL1, CTL4, CTL10), the domain average total runoff is about 17% of the rainfall (2-year runoff ratio = 0.17), while ET is 84%. The combined total of ET and runoff is slightly higher than rainfall over the two years, which is balanced by 315 a slight decrease in soil moisture. The inclusion of lateral flow increases the 2-year cumulative ET by 2 to 4%. As ET is a substantial component of the surface water balance, this translates to larger changes of -11 to -22% in total runoff (2-year runoff ratio with lateral flow = 0.13 to 0.15). The changes in ET, soil moisture and total runoff with the inclusion of lateral flow are summarised in Table 3.

**Table 3: Changes in cumulative domain average surface water components over 2-years from 2015-12 to 2017-11 due to the inclusion of lateral flow.**

| Experiment | ET change in mm (in %) | Soil moisture change in mm (in %) | Total runoff change in mm (in %) |
|---|---|---|---|
| LAT1-100 minus CTL1 | 43 (3%) | 6 (32%) | -49 (-19%) |
| LAT1-250 minus CTL1 | 25 (2%) | 4 (24%) | -29 (-11%) |
| LAT4-100 minus CTL4 | 54 (4%) | 4 (21%) | -58 (-22%) |
| LAT4-250 minus CTL4 | 45 (4%) | 3 (18%) | -48 (-19%) |
| LAT10-100 minus CTL10 | 49 (4%) | 3 (17%) | -52 (-20%) |
| LAT10-250 minus CTL10 | 44 (3%) | 2 (12%) | -46 (-18%) |

### 3.2.2 Consistency with observations

The higher ET near the river channels in the model simulations including lateral flow are consistent with patterns from the 325 high-resolution CMRSET dataset. Figure 6 compares the spatial anomalies in ET from the high-resolution 1-km LSM simulations (CTL1 and LAT1-250) with those from the CMRSET data. The spatial ET anomalies from CMRSET indicate the presence of higher ET near the channels in the downstream areas of Murrumbidgee and Murray Riverina basins in DJF, MAM and SON (Fig. 6a, 6c, 6d). The anomalies in the LAT1-250 simulations are also positive near the channels in these seasons (Fig. 6i, 6j, 6l). The CTL1 simulations without lateral flow do not show this pattern (Fig. 6e-h). However, the modelled spatial 330 anomalies in the LAT1-250 simulations are larger in magnitude than the CMRSET anomalies, especially in DJF and SON.

The CMRSET spatial anomalies near the channels also appear to be more spatially diffused in contrast to the LAT1-250 simulations. This could be because processes such as seepage from the channels to the soil column and re-infiltration of over bank inundated water are currently not represented in WRF-Hydro. Nevertheless, the higher ET near major river channels in the satellite-based CMRSET product qualitatively supports the patterns in our high-resolution simulations including lateral flow.

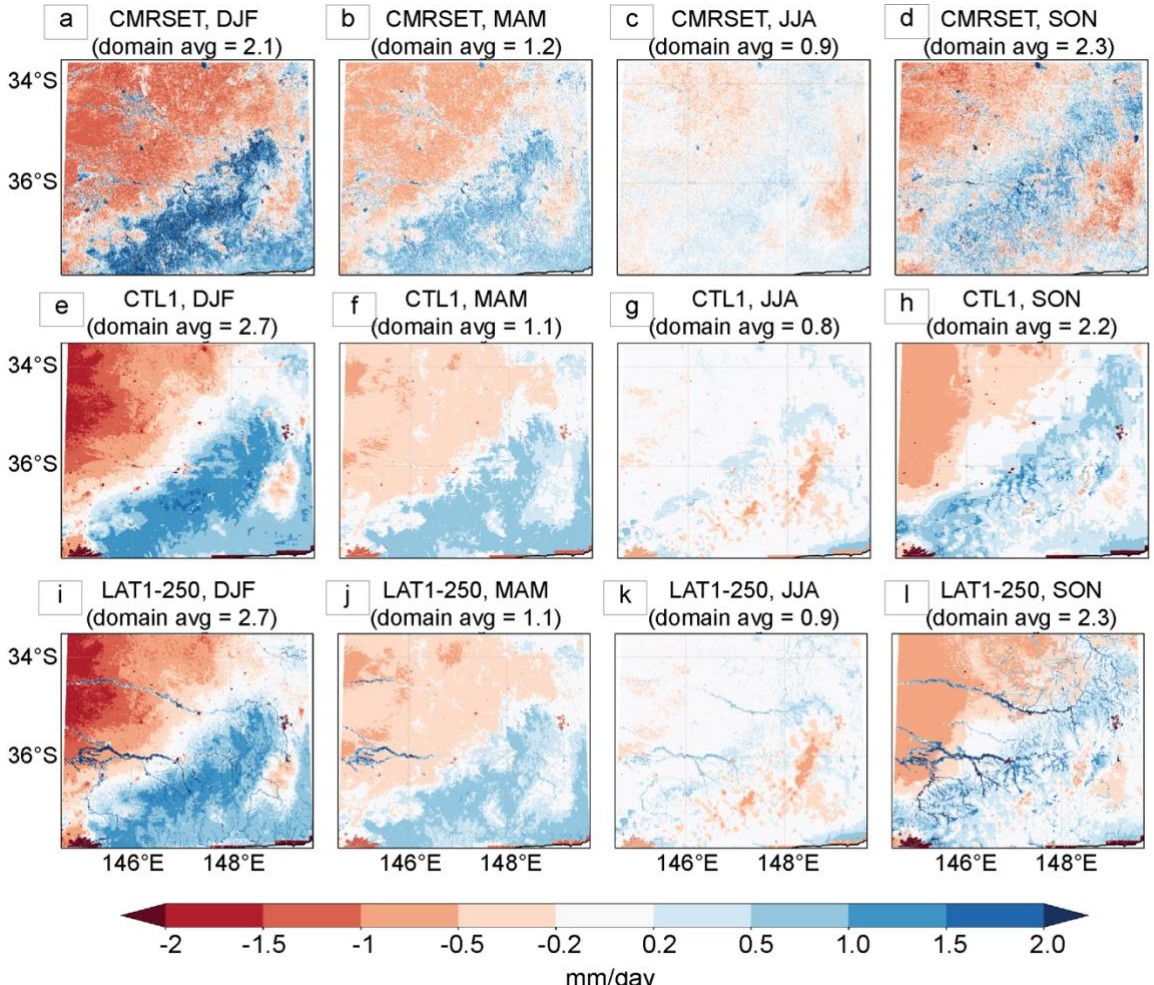

**Figure 6: Comparison of the spatial anomalies in seasonal ET from the corresponding domain averages (in mm/day) from (a-d) CMRSET satellite-based data (e-h) CTL1 (i-l) LAT1-250 model simulations. The domain averages from which the anomalies are calculated is shown above each panel (in mm/day).**

### 3.2.3 Larger impact in downstream basins

We quantify the area average changes in ET in basins that exhibit large changes with inclusion of lateral flow in Fig. 7, the Upper (includes Upper Murray and Kiewa), Ovens and Murray Riverina basins (outlined in Fig. 1). The Upper and Ovens basins receive substantial amount of rainfall in the cool season (Apr-Sep). The Upper basin is characterised by steep

topographic variation of more than 10 m/km (Fig. 1), while the Ovens basin has gentler topographic variation. Murray Riverina,
located downstream of the Upper and Ovens basins, receives lower rainfall and has flatter topography. Figure 7b-c shows the basin average changes in ET with inclusion of lateral flow at different resolutions. In the Upper basins, the 1-km resolution simulations exhibit neutral to slightly negative changes in ET, indicating that the drier ridges seen in the mean seasonal ET (Fig. 4 and 5) dominate the basin average signal at this resolution. In contrast, the coarser resolution simulations exhibit positive changes in this basin. In the Ovens basin, the changes in ET are generally positive at all resolutions. However, the changes in
the 1-km resolution runs are much smaller in magnitude than in the coarser LSM resolutions. In Murray Riverina, the 1-km and 4-km resolution simulations exhibit the largest influence due to lateral flow. The changes in this basin amount to ~1 mm/day (50%) after the wet cool season of 2016, and the changes in ET in this region persist through to April 2017. Considering the two different routing resolutions, a 100 m routing grid tends to increase the changes in ET compared to the 250 m grid at all LSM resolutions in these basins. LSMs are known to systematically vary and generally underestimate ET
during dry down periods when compared to flux tower observations (Ukkola et al., 2016). Our results show that lateral flow could be one of the processes that contribute to dry down biases in LSMs, especially in regions that are downstream of topography. The increase in ET due to lateral flow results in cooler maximum air temperatures in Summer. In simulations at 4 and 1 km resolutions, the Murray Riverina basin average monthly maximum temperatures from Dec 2016 to Jan 2017 is about 0.3 degrees cooler in simulations including lateral flow.


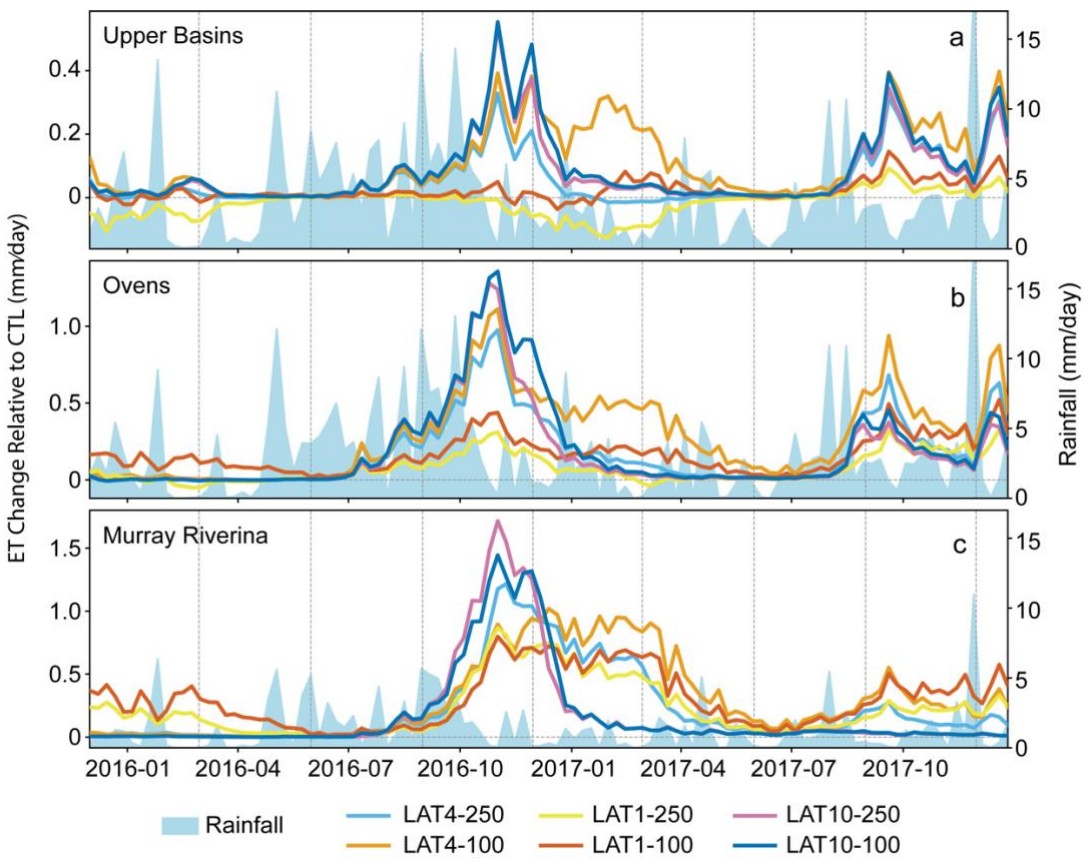

**Figure 7: Timeseries of basin average rainfall (filled line plot) and changes in ET (in mm/day; lines) with inclusion of lateral flow in the Upper, Ovens, and Murray Riverina Basins.**

## 4 Conclusions

We used standalone LSM simulations at different spatial resolutions to quantify the influence of lateral flow on ET and surface water partitioning in a topographically diverse region in southeast Australia. LSM simulations at resolutions of 4- and 1-km show lateral flow induced increases in ET near major channels in spring and summer. Similar spatial anomalies exist in high-resolution observations. Simulations at 10-km resolution exhibit different spatial pattern of changes with the ET increases confined to upstream locations. The high-resolution 1-km simulations also exhibit widespread drying at the upstream ridges, which are absent at 4- and 10-km resolutions. The dry ridges and wet valleys in our high-resolution (1-km) simulations are consistent with understanding from hillslope scale critical zone studies (Fan et al., 2019), and with results from high-resolution simulations in a test watershed in the U.S. (Qiu et al., 2024). The domain average ET increases due to lateral flow from our

simulations are small (2 to 4%), consistent with prior assessments (Chaney et al., 2021; Lahmers et al., 2020). We find that the changes in surface water partitioning reduces domain average runoff by between -11 and -22%.

Our results have implications for the utility of LSMs in water management applications. In southeast Australia, rainfall-runoff relationships are already changing, and the observed changes are thought to be linked to changes in evapotranspiration, vegetation, and vadose zone recharge processes (Fowler et al., 2022; Peterson et al., 2021). These shifts are challenging to simulate in conceptual rainfall-runoff models that are currently used to develop streamflow projections for management decision making (Gardiya Weligamage et al., 2023). LSMs could develop into a viable alternative for these types of application. In our simulations, lateral flow modulates runoff by changing the surface water partitioning between evaporation and runoff, suggesting that including this process may enhance the ability of LSMs to model hydrological shifts.

Our results also have implications for modelling droughts and future water availability. Our simulations show largest increases in ET in the downstream Murray Riverina region; after a wet winter lateral flow lengthens the ET dry down by more than 3 months in this region. This suggests that incorporating lateral flow processes can reduce long standing LSM deficiencies in simulating seasonal droughts (Ukkola et al., 2016) in areas where flow convergence effects are dominant. Further, assessments at finer spatial resolutions report that lateral water movement has a dominant influence on ecohydrological outcomes both during climate extremes in the past (Mastrotheodoros et al., 2020; Norton et al., 2022), and in a warming climate (Stephens et al., 2022). While the impact of lateral flow on domain average ET is small, the pattern of dry ridges and wet valleys indicate the potential for local differences in drought onset, with ridges entering periods of soil moisture droughts faster than the valleys. These spatial patterns have implications for coupled climate modelling as idealised land-atmosphere simulations in literature report that such spatial heterogeneities can feedback to induce secondary atmospheric circulations and affect cloud structure (Sakaguchi et al., 2022).

Our results show that the influence of lateral flow on soil moisture and land-atmosphere fluxes depends on model spatial resolution. In simulations at 10-km resolution, the spatial pattern of changes due to lateral flow differ from the high-resolution simulations and observations. This suggests that it may not be essential to include lateral flow in all current generation of regional climate projections which are at resolutions of order 10 km, including the CMIP6-based CORDEX Australasia projections (Grose et al., 2023). Our results are specific to southeast Australia, and this conclusion needs to be tested in other regions as the degree to which this conclusion is transferable is not clear. However, the next generation of climate modelling is moving towards km-scale resolutions (Wedi et al., 2020) and the Earth Virtualization Engine (EVE) (Stevens et al., 2024) aims to provide information at ~1 km scale granularity, globally to aid decision making. Our results suggest that the influence of lateral flow cannot be ignored at these higher spatial resolutions.

Our simulations have limitations that indicate two main avenues of future work required to improve the representation of lateral flow in LSMs in southeast Australia. First, our simulations are sensitive to soil information, indicating the potential for improvement through modification of soil types and parameters. Such sensitivities to input datasets (Yang et al. 2021) and model parameters (Chaney et al, 2021, Zhang et al. 2024) are also noted in other regional studies. Combining field measurements with remotely sensed soil information have been shown to bring benefits in the U.S. (Xu et al., 2023) and may

be studied in future work in Australia. Exploring the transferability and utility of Australian regional soil information developed for water accounting (Dutta et al., 2017; Viney et al., 2015) to regional land surface modelling may also be worthwhile. Second, more accurate representation of processes including baseflow and the feedback from the channels to the soil columns may be relevant in arid and semi-arid regions, such as southeast Australia. These processes are not represented in our simulations. Our calibration results suggest that spatially varying baseflow may improve lateral flow representation in our domain. Incorporating soil layers deeper than the 2-m depth modelled in standard Noah-MP have reduced surface flux biases in some cases in other semi-arid locations (Gochis et al., 2010; Barlage et al., 2015) and may be explored in future work. Incorporating channel seepages have improved streamflow simulations in semi-arid Arizona (Lahmers et al., 2021), but the feedback of this process to the soil column has not been modelled yet.

In summary, our results suggest that existing coarse resolution simulations of surface fluxes over southeast Australia are not sensitive to the omission of lateral flows. However, at high spatial resolution, these fluxes are sensitive and the additional complexity of including lateral flows becomes important. There are existing methods of capturing lateral flows in land surface models so our conclusion is not an impediment to using high-resolution models, but ensuring that these components are tested and implemented in modelling systems remains a priority.

**Appendix A: Additional Figures**

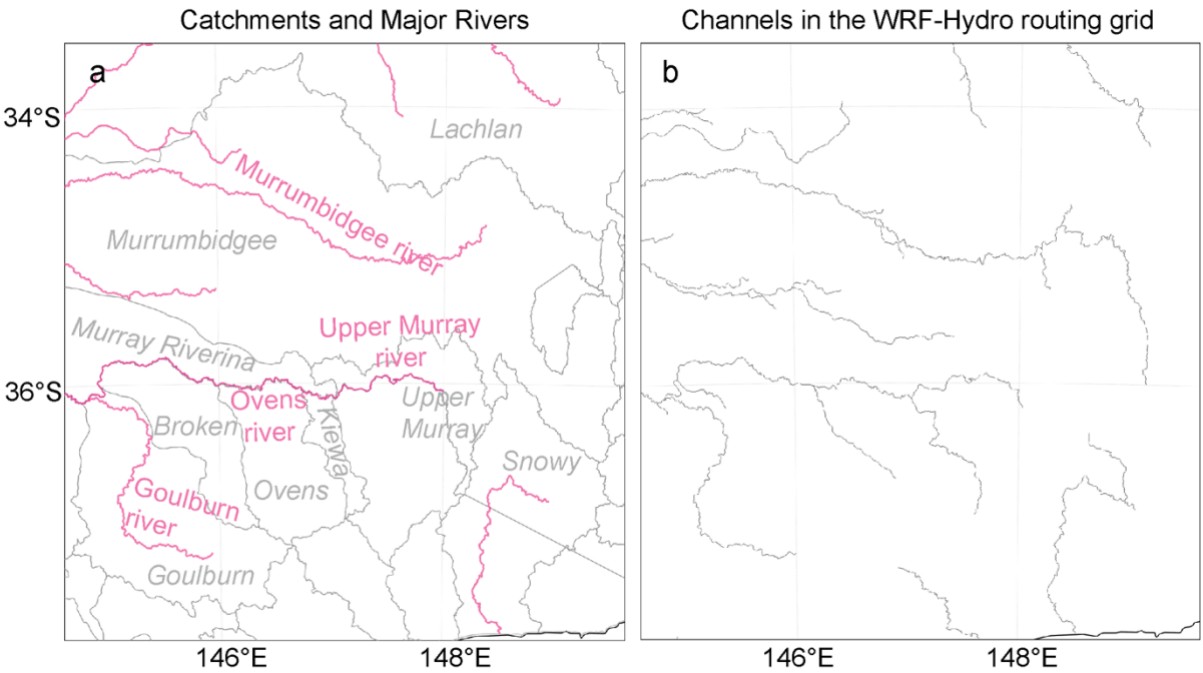

**Figure A1: (a) Surface water catchments (grey outlines and text) based on data from FAO AQUASTAT (https://data.apps.fao.org/catalog/dataset/6a53d768-1e20-46ea-92a8-c4040286057d) and major rivers (pink lines and text) based on**

data from Geoscience Australia (https://pid.geoscience.gov.au/dataset/ga/42343), and (b) higher order channels (stream order of four or higher) used in the WRF-Hydro model.

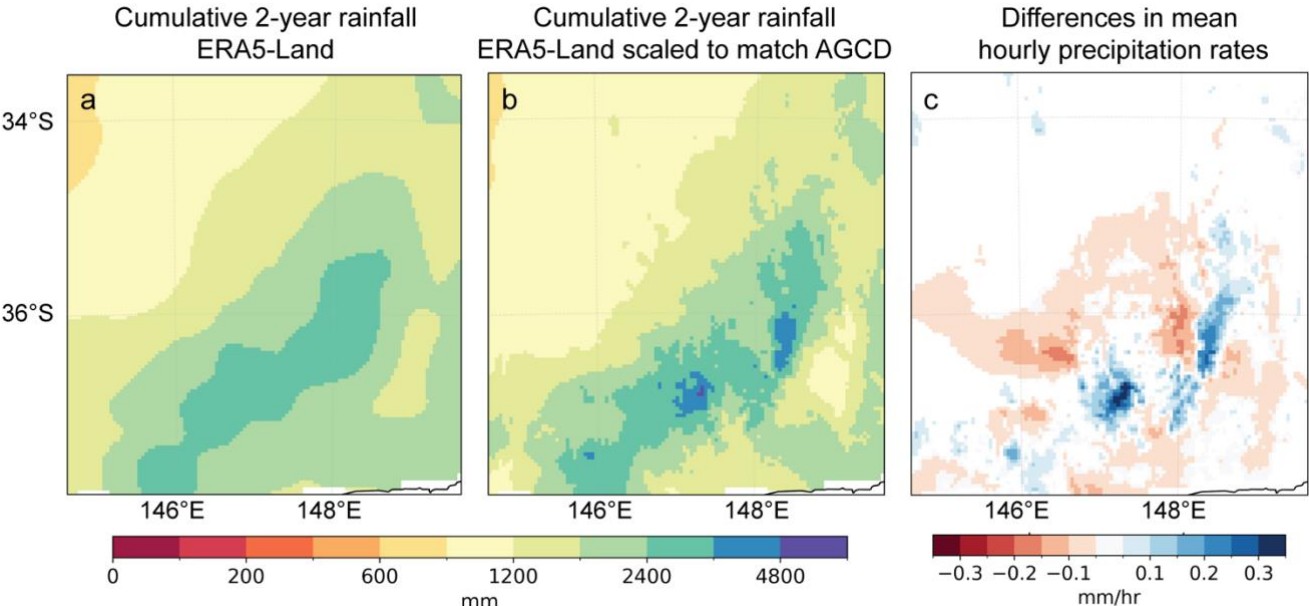

Figure A2: (a) Cumulative 2-year rainfall during 2016-17 from the ERA5-Land dataset, (b) Cumulative 2-year rainfall during 2016-17 from ERA5-Land scaled to match the AGCD dataset, (c) mean differences in hourly precipitation rates in 2016-17. Only grids that show significant differences (at 5% significance level) are shaded in panel (c).

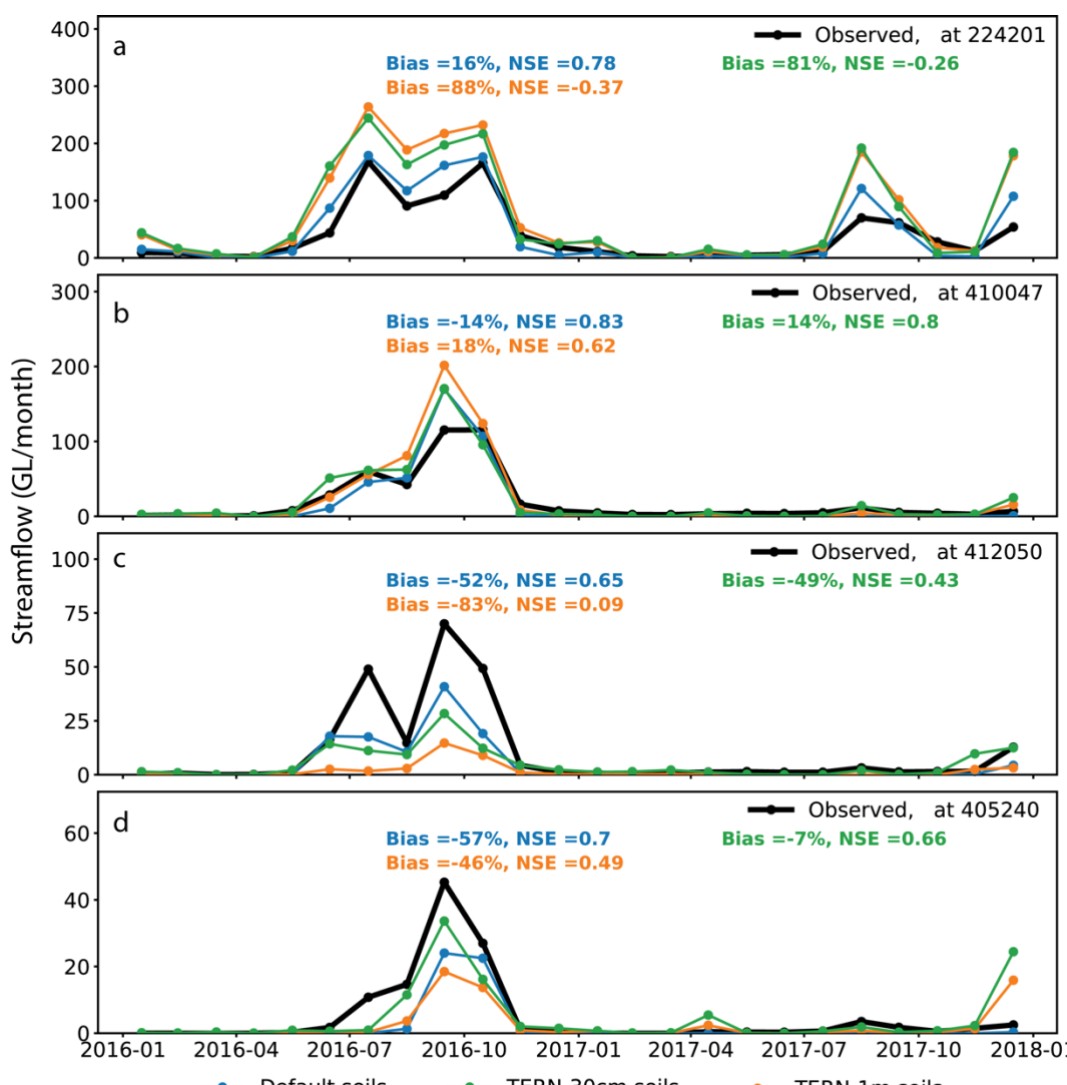

Figure A3: Sensitivity of streamflow to soil data: simulations using the default soil data, and with soil classes specified based on 1-m and 30-cm depths of the TERN soil data. Simulated monthly streamflow using different soil data (coloured lines) compared to observed streamflow (black lines) at the gauge locations (in GL/month) in simulations with an LSM resolution 4 km and routing resolution 250 m (LAT4-250). All simulations are calibrated to streamflow as described in the main text. The biases and NSE of the simulations are indicated by the numbers in the corresponding colours in each panel.

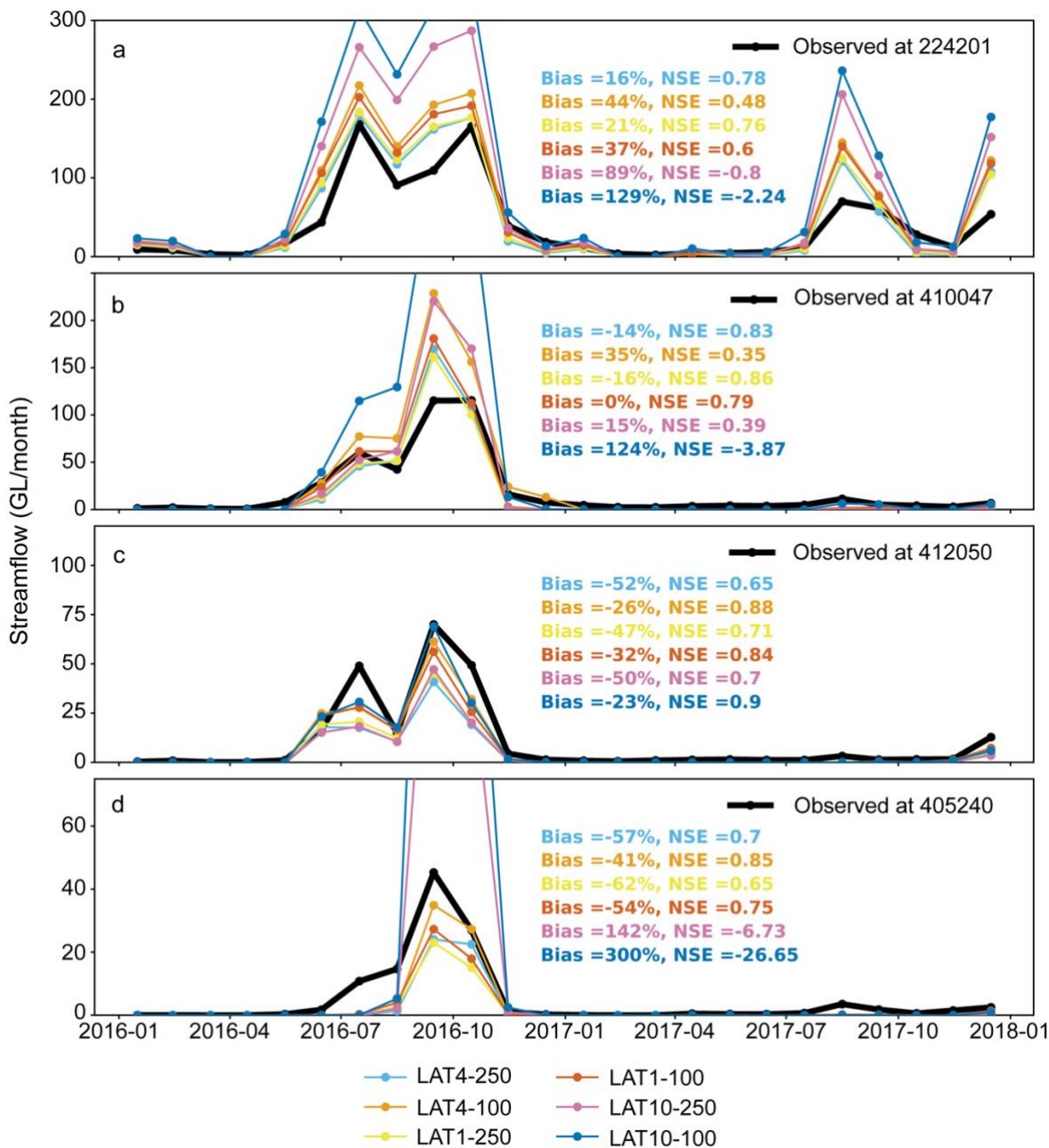

440

**Figure A4: Simulated cumulative monthly streamflow using the calibrated parameter values compared to observed streamflow (in GL/month) at the gauge locations in simulations at different resolutions.**

**Table A1. Noah-MP parameterisation options used for the simulations**

| Noah-MP physics parameterisation | Option selected |
|---|---|
| Dynamic vegetation | 4 – Leaf area index/stem area index from lookup table; maximum vegetation fraction from climatology |
| Stomatal resistance | 1 – Ball-Berry formulation |
| Soil moisture reduction for stomatal resistance controlling | 1 – Similar to original Noah based on soil moisture |
| Runoff | 3 – Infiltration excess surface runoff and free drainage subsurface runoff |
| Surface exchange coefficient | 1 - Monin -Obukhov similarity |
| Frozen soil | 1 – Hydraulic properties from total soil water and ice (Niu and Yang, 2006) |
| Supercooled liquid water in frozen soil | 1 – General form of freezing-point depression equation (Niu and Yang, 2006) |
| Radiative transfer | 3 - Two-stream approximation applied to vegetated fraction |
| Snow albedo | 2 – From land surface scheme CLASS |
| Partitioning precipitation into rainfall and snowfall | 1 – Formulation as in Jordan (1991) |
| Lower boundary condition of soil temperature | 2 – Fixed lowest soil temperature from input |
| Temperature time scheme | 3 – Semi-implicit but split by snow fraction |
| Surface resistance to evaporation/sublimation | 4 – Sakaguchi and Zeng (2009) for non-snow and separate snow resistance for snow fraction |

**Table A2. Domain average water cycle terms accumulated over a 2-year period from 2015-12 to 2017-11 in the simulations. Negative soil moisture changes indicate a loss of moisture from the 2-m soil column over the 2-year period.**

| Variable | Simulation | | | | | | | | |
|---|---|---|---|---|---|---|---|---|---|
| | CTL1 | LAT1-100 | LAT1-250 | CTL4 | LAT4-100 | LAT4-250 | CTL10 | LAT10-100 | LAT10-250 |
| Precipitation, $P$ (mm) | 1504.4 | 1504.4 | 1504.4 | 1504.7 | 1504.7 | 1504.7 | 1503.1 | 1503.1 | 1503.1 |
| $ET$ (mm) | 1264.3 | 1307.7 | 1289.1 | 1262.9 | 1317.2 | 1308.1 | 1260.8 | 1309.6 | 1304.8 |
| Soil moisture change, $SM$ (mm) | -17.9 | -12.2 | -13.5 | -17.9 | -14.2 | -14.6 | -17.9 | -15.0 | -15.7 |
| Surface runoff, $sfcrnoff$ (mm) | 46.2 | | | 46.5 | | | 47.5 | | |
| Underground runoff, $udgrnoff$ (mm) | 211.8 | | | 213.2 | | | 212.7 | | |
| (a) Total runoff ($sfcrnoff$ + $udgrnoff$), Ro (mm) | 258.0 | | | 259.7 | | | 260.2 | | |
| (b) Total runoff ($P – ET – SM$), $Ro$ (mm) | 258.0 | 208.9 | 228.8 | 259.7 | 201.7 | 211.2 | 260.2 | 208.5 | 214.0 |
| Water balance closure error, $P – ET – SM – sfcrnoff - udgrnoff$ (mm) | 0.0 | | | 0.0 | | | 0.0 | | |
| Runoff ratio ($Ro/P$) | 0.17 | 0.14 | 0.15 | 0.17 | 0.13 | 0.14 | 0.17 | 0.14 | 0.14 |

**Code availability**

The code is available at https://github.com/anjanadevanand/LSM_lateral_flow_SEAus.

**Data availability**

The processed model outputs of evapotranspiration and streamflow used for analyses will be published in zenodo repository 10.5281/zenodo.13836371 on acceptance of the paper. The DOLCE v3 data is available from https://researchdata.edu.au/derived-optimal-linear-dolce-v30/1697055. The CMRSET version 2.2 data is available from https://portal.tern.org.au/metadata/TERN/9fefa68b-dbed-4c20-88db-a9429fb4ba97. The gauged streamflow data is available from http://www.bom.gov.au/water/hrs/.

**Author contribution**

Anjana Devanand, Jason P. Evans and Andy J. Pitman conceptualised the study and designed the experiments. Anjana Devanand performed the simulations with inputs from Sujan Pal, David Gochis and Kevin Sampson. Anjana Devanand performed the analyses and prepared the figures with inputs from Jason P. Evans and Andy J. Pitman. Anjana Devanand wrote the paper, and all authors contributed to editing and review.

**Competing interests**

The authors declare that they have no conflict of interest.

**Acknowledgments**

We acknowledge funding from the ARC Centre of Excellence for Climate Extremes (CE170100023). This research was undertaken with the assistance of resources and services from the National Computational Infrastructure (NCI), which is supported by the Australian Government.

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
