# Peer review of "The influence of lateral flow on land surface fluxes in southeast Australia varies with model resolution"

_EGUsphere, 2024_

## Author Comment (AC1)

**Referee 1**

**General Comments:**

**This is an interesting paper that investigates the importance of lateral transfers of water and its effects on energy (mainly ET) within semiarid and complex terrain locations of Southeastern Australia. While not novel in addition of any extra model physics, it is an important addition to the scientific community that investigates land surface models and the bridge between hydrologic models. That being said, there are a number of comments and concerns that I have had while reading through this text. Specifically, I am concerned about the method that was used to bias correct precipitation inputs and the calibration period (which was only a length of 45 days). I would implore the authors to better support these decisions within the manuscript. Based on this initial draft, I would rate this as Fair on Scientific Significance, Good on scientific quality (mainly needing more justification), and Good on presentation quality and suggest major revisions to address comments below:**

We thank the referee for the encouraging comments, and suggestions to justify the decisions around bias correction of precipitation and the calibration period. We have given a point-by-point response to all the comments, describing the changes that will be made to the revised manuscript to incorporate them. We believe that the revisions fully satisfy the referee's concerns.

Referee comments are shown in bold. Author responses are shown in plain text.

**Major Comments (in order of where they are in the text, not in order of importance):**

1. **Paragraph beginning on Line 103: Within this paragraph, the authors explain different overland flow and sub surface flows and how these and cannot feedback into soil water and energy fluxes. Please explicitly state what is meant here by lateral transfers (e.g. case 2b), and if the subsurface flow is still being parametrized despite the baseflow package being turnoff due to calibration. This paragraph is critical to understanding the scientific set-up of the study, and thus needs to be crystal clear.**

We thank the referee for the question. The lateral transfer we refer to includes both overland and subsurface flow (despite the baseflow being turned off), and these processes feedback to affect the LSM states in our 'LAT' simulations. We will revise the description of the scientific set-up for clarity as below:

"WRF-Hydro has the capability to simulate overland, shallow subsurface, and channel flows. The subsurface flow is simulated on the LSM grid, while overland and channel flows are simulated on the fine grid. The subsurface lateral flux in the saturated portion of the soil column is calculated based on hydraulic gradients using the method documented in Wigmosta et al. (1994) and Wigmosta and Lettenmaier (1999), implemented in the Distributed Hydrology and Soil Vegetation Model (Gochis et al., 2020). Overland and channel flow is calculated using the diffusive wave formulation, using Manning's equation as the resistance formulation (Gochis et al., 2020). In the simulations presented in this paper, overland and shallow sub-surface flows are modelled, and these processes feedback to influence soil moisture and surface fluxes in the LSM. The water that flows into the channel grids is routed downstream to model streamflow and does not feedback to affect the LSM soil moisture. In other words, channel leakages are not represented in the model. The WRF-Hydro modelling system also has the functionality to specify a conceptual representation of baseflow by passing the underground runoff from the LSM directly into the channel network, but this is not used in the simulations presented in this paper. We calibrate model parameters to match streamflow at selected gauges. Based on preliminary calibration results, the baseflow representation is turned off in our simulations. It is worth noting that lateral saturated sub-surface flow is being modelled in our simulations as described above, despite baseflow being turned off."

2. **Please expand, especially on the precipitation, the bias correction used. Is the idea here that you take a monthly accumulated rainfall at each grid cell from ERA5 land and the Australian Gridded Climate Data (AGCD) and scale each month to directly match the Australian Gridded climate data set? How does this effect the hourly precipitation rates? Infiltration rates will be highly sensitive to the hourly rainfall rates, so ensuring this is clearly explained is critical. See "*Sampson AA, Wright DB, Stewart RD, LoBue AC. The role of rainfall temporal and spatial averaging in seasonal simulations of the terrestrial water balance. Hydrological*"**

*Processes. 2020; 34: 2531–2542"* **for evidence showing that at hourly scale, rainfall is driving much of the uncertainty of infiltration, not necessarily the soil parameters (though these are very much still important).**

Yes, the referee is correct about the idea of bias correction used. We will clarify this in the revised manuscript. We thank the referee for raising the point about the changes in precipitation rates and its effect on infiltration. We agree that this is important to clarify this in the paper. In the revised manuscript, we will elaborate on the necessity for bias correction and document the changes in precipitation rates induced by the monthly scaling correction as below:

"The monthly accumulated precipitation and mean temperature from ERA5-Land reanalysis are scaled to match the AGCD observations for the corresponding month at each grid. This correction is performed as AGCD is a high-quality historical data developed by applying topography resolving analyses methods to in situ observations (Jones et al., 2009) and is expected to be more accurate than reanalysis data. The corrected forcing data would thus match the AGCD closely at monthly timescales, while using the sub daily pattern of variation in the ERA5-Land reanalysis. Compared to ERA5-Land, the AGCD precipitation exhibits higher spatial variation primarily over the mountainous areas in the domain (Fig. A2), and these differences can have substantial effect on infiltration rates in land simulations (Sampson et al., 2020). Application of the monthly scaling correction induces localised increases and decreases in hourly precipitation rates from ERA5-Land as shown in Fig. A2(c)."

[Figure]

Figure A2: (a) Cumulative 2-year rainfall during 2016-17 from the ERA5-Land dataset, (b) Cumulative 2-year rainfall during 2016-17 from ERA5-Land scaled to match the AGCD dataset, (c) mean differences in hourly precipitation rates in 2016-17. Only grids that show significant differences (at 5% significance level) are shaded in panel (c).

3. **Please provide information on the 45 day period that was used to calibrate the model. Were these high flow days? Were they low flow days? Why was such a small period of time (45 days aggregated by 3 days is 15 data points to calibrate on). More justification is needed. Specifically, why does it make sense here to calibrate to 3 daily flow (assuming accumulated), when the comparisons will be on monthly flow (accumulated as well?) I understand calibration is tricky, and am not advocating for the authors to do more work, but do think that justifying this choice somehow is necessary.**

We thank the referee for raising this important point. Calibration is indeed tricky, and preliminary work was undertaken to arrive at the calibration settings documented in the manuscript. We will include the following information in the revised manuscript to provide justification for the choices.

"Calibration of WRF-Hydro is computationally intensive and involves choices that may be aligned to the purpose of the simulations. Here we study the influence of lateral flow on seasonal timescales and hence the main purpose of calibration is to obtain better streamflow outcomes on monthly to seasonal timescales, rather that improved simulations of daily scale streamflow events. The streamflow in the domain primarily occurs in the cool season (May to October), and model simulations using default parameter values exhibit biases during these high flow months (Fig. 2). However, preliminary results showed that event-based calibration to high flow days did not translate to improved monthly flows indicating that it is necessary to use a period at least of the order of a month, which includes both high and low flow days at the four gauges for calibration. As a 45-day period is computationally feasible, and yields reasonable outcomes at monthly to seasonal timescales, this length of time is chosen for calibration. The daily streamflow data is smoothed by aggregating to 3-day flows to dampen the effect of individual high flow days."

4. **Figure 7: Please add a ET Change Relative to CTL label on the y axis. Also please ensure the labels are all correct (CTL1-250 doesn't exist in this study).**

We thank the referee for the comment on Figure 7, and for noting the error in the legend. We will address these in the revised version of Figure 7 shown below.

[Figure]

Figure 7: Timeseries of basin average rainfall (filled line plot) and changes in ET (in mm/day; lines) with inclusion of lateral flow in the Upper, Ovens, and Murray Riverina Basins.

**Minor comments (in order of where they are in the text, not in order of importance):**

5. **Great introduction! I would contend that there could be a nod to some of the work that is being done in the Urban world with lateral transfers (understanding that this is not the scope of this paper, but is an important emerging area where hydrologic processes are just as important and often overlooked in LSMs). I would think a clear location to add would be in the paragraph starting on line 70.**

We thank the referee for the suggestion and will include a mention of the work on lateral flow in urban areas in the introduction.

"….These studies report that while increases in regional mean soil moisture and evapotranspiration (ET) induced by lateral flow are small, the changes are spatially heterogenous (Chaney et al., 2021; Fersch et al., 2020; Lahmers et al., 2020; Qiu et al., 2024; Zhang et al., 2024). In urban areas, fine scale heterogeneity of impervious areas and open spaces may induce substantial changes in the surface energy balance (Alexander et

al., 2024; Reyes et al., 2016). On regional scales, including lateral ground water flow is reported to increase the proportion of ET from transpiration over the United States (Maxwell and Condon, 2016)….."

References added:

Alexander, G. A., Voter, C. B., Wright, D. B., and Loheide II, S. P.: Urban Ecohydrology: Accounting for Sub-Grid Lateral Water and Energy Transfers in a Land Surface Model, Water Resources Research, 60, e2023WR035511, https://doi.org/10.1029/2023WR035511, 2024.

Reyes, B., Maxwell, R. M., and Hogue, T. S.: Impact of lateral flow and spatial scaling on the simulation of semi-arid urban land surfaces in an integrated hydrologic and land surface model, Hydrological Processes, 30, 1192–1207, https://doi.org/10.1002/hyp.10683, 2016.

6. **Figure 1b): please change the outline color of the Upper Basins, Ovens, and Murray Riverina to something that isn't blue. These are currently will be difficult to differentiate given the light blue color used for the terrain height being for much of the lowlands.**

Thanks for the suggestion. We will revise Figure 1 as shown below to incorporate this comment. We will also include channels in the figure as suggested by Referee 2.

[Figure]

Figure 1: (a) The WRF-Hydro model domain in southeast Australia. (b) Features in the domain. Background shading indicates topography, and the surface water catchments (black outlines) that drain into the streamflow gauges (blue triangles) used for calibration are marked on the map. The basins outlined in red (Upper Basins, Ovens, and Murray Riverina) are used to analyse the influence of lateral flow in basins with varying topographic characteristics. The network of major rivers (blue lines) based on data from Geoscience Australia (https://pid.geoscience.gov.au/dataset/ga/42343) are shown in panel b.

7. **Somewhere within the manuscript or within an appendix, please list the specific choices made for the Noah-MP LSM in terms of physics schemes used. While out of the scope of this paper, these have a very clear influence on the results of the model, and should be listed.**

We thank the referee for raising this. We will add Appendix Table A1 listing the Noah-MP parameterisation options used for the simulations in the revised manuscript.

Table A1. Noah-MP parameterisation options used for the simulations

| Noah-MP physics parameterisation | Option selected |
| --- | --- |
| Dynamic vegetation | 4 – Leaf area index/stem area index from lookup table; maximum vegetation fraction from climatology |
| Stomatal resistance | 1 – Ball-Berry formulation |
| Soil moisture reduction for stomatal resistance controlling | 1 – Similar to original Noah based on soil moisture |
| Runoff | 3 – Infiltration excess surface runoff and free drainage subsurface runoff |
| Surface exchange coefficient | 1 - Monin -Obukhov similarity |
| Frozen soil | 1 – Hydraulic properties from total soil water and ice (Niu and Yang, 2006) |
| Supercooled liquid water in frozen soil | 1 – General form of freezing-point depression equation (Niu and Yang, 2006) |
| Radiative transfer | 3 - Two-stream approximation applied to vegetated fraction |
| Snow albedo | 2 – From land surface scheme CLASS |
| Partitioning precipitation into rainfall and snowfall | 1 – Formulation as in Jordan (1991) |
| Lower boundary condition of soil temperature | 2 – Fixed lowest soil temperature from input |
| Temperature time scheme | 3 – Semi-implicit but split by snow fraction |
| Surface resistance to evaporation/sublimation | 4 – Sakaguchi and Zeng (2009) for non-snow and separate snow resistance for snow fraction |

8. **Line 115: The "eight seasons" seems to be obfuscating the amount of analysis done. Why not just "2 years of results, broken into individual seasons" or something similar?**

Thanks for seeking clarification. The results are presented for four individual seasons from two years of simulations (Figure 4, Figure 5, Figure 6), rather than eight seasons. The wording of this sentence is likely confusing, and we will revise it to convey this clearly as: "We simulate years 2013 to 2017 and analyse the changes in ET during 2015-12 to 2017-11 broken into individual seasons, discarding the first ~3 years as spin-up."

9. **I am being pedantic here, but please define monthly streamflow; is this an average or an accumulation over the whole month? I assume it is an accumulation, but could not find it confirmed in the text.**

Thanks for seeking clarification. It is cumulative monthly streamflow, and we will clarify it in in the text in Section 3.1.1, and in the captions of Figure 2 and revised Figure A4.

10. **Figure A3 panel a: why is there a single dot in the middle of the panel behind all of the text. Is this an erroneous plot? Also, please move the Bias and NSE results so that they do not overlap any of the lines. It is hard to read!**

We thank the referee for spotting this. The single dot behind the text was erroneous and we have removed it. The figure has also been updated so that the text listing the Bias and NSE results do not overlap any of the lines. The revised figure (Fig. A4 in the revised manuscript) is shown below.

[Figure]

Figure A4: Simulated streamflow using the calibrated parameter values compared to observed streamflow (in GL/month) at the gauge locations in simulations at different resolutions.

11. **Please revise "*The simulated timeseries of ET are within the range from the DOLCE product most of the time, except in 6 out of 24 months where the simulations are slightly outside this range.*" 25% of the time being outside of the uncertainty range is a pretty significant amount to be outside of the uncertainty estimates.**

Thanks for the suggestion. We will revise this sentence to state that: "The simulated timeseries of ET are within the range from the DOLCE product 75% of the time, and slightly outside this range the rest of the time (6 out of 24 months)."

12. **For ET in Figure 3: Please specify whether or not this is over the full domain in Figure 1b or just within the sub-catchments of interest somewhere in the text.**

This figure shows the ET averaged over the full domain in Figure 1b. We will clarify this in the text as: "We compare the simulated domain average monthly ET (domain shown in Fig. 1b) during 2016-17 with…"

---

## Author Comment (AC2)

**Referee 2**

**The results of this paper are significant for the hydrologic modeling and LSM communities, as they demonstrate the impacts lateral flow in LSMs and the possible implications for atmospheric fluxes. The methods of this manuscript are generally strong, and I anticipate that the results of this work could have implications for the hydrometeorological community.**

**However, I have some technical concerns on the methods that I ask the authors to clarify in order for this manuscript to be accepted.**

We thank the referee for the encouraging comments, and for seeking clarification on some of the technical aspects. We have given a point-by-point response to all the comments, describing the changes that will be made to the revised manuscript to incorporate them. We believe that the revisions fully satisfy the referee's concerns.

Referee comments are shown in bold. Author responses are shown in plain text.

**Major Comments:**

1. **Section 2.1, Lines 100-110: I appreciate this discussion on the significance of horizontal routing with LSMs. I also recommend discussing the impacts of LSM depth, as the standard 2-m Noah-MP depth often does not capture groundwater processes and has resulted in dry biases with ET in some regions.**

We thank the referee for raising this and agree that the 2-m soil depth may contribute to LSM biases. In the revised manuscript, we will elaborate on this point as below.

"Here we use the standard Noah-MP LSM which has a constant soil depth of 2 m with vertically homogeneous soil parameters. This formulation can contribute to biases in runoff and evapotranspiration, which may be ameliorated by incorporating variable and higher soil depths, groundwater processes, and vertical soil heterogeneity (Gochis et al., 2010; Barlage et al., 2015; Wu et al., 2021; Yimam et al., 2025) in the modelling framework. These aspects are outside the scope of the work presented in this study and have not been explored here."

References added:

Barlage, M., Tewari, M., Chen, F. *et al.* The effect of groundwater interaction in North American regional climate simulations with WRF/Noah-MP. *Climatic Change* **129**, 485–498 (2015). https://doi.org/10.1007/s10584-014-1308-8

Gochis, D. J., Vivoni, E. R., and Watts, C. J.: The impact of soil depth on land surface energy and water fluxes in the North American Monsoon region, Journal of Arid Environments, 74, 564–571, https://doi.org/10.1016/j.jaridenv.2009.11.002, 2010.

Wu, W.-Y., Yang, Z.-L., and Barlage, M.: The Impact of Noah-MP Physical Parameterizations on Modeling Water Availability during Droughts in the Texas–Gulf Region, Journal of Hydrometeorology, 22, 1221–1233, https://doi.org/10.1175/JHM-D-20-0189.1, 2021.

Yimam, Y. T., Neely, H. L., Morgan, C. L. S., Kishné, A., Gross, J., and Gochis, D.: Evaluation of Noah-MP performance with available soil information for vertically heterogenous soils, Agrosystems, Geosciences & Environment, 8, e70048, https://doi.org/10.1002/agg2.70048, 2025.

2. **Section 2.2 and 2.3, Calibration: Please clarify why the authors use 3-day averaging for streamflow validation and calibration? This could likely underestimate major surface runoff events leading to flash flooding. Furthermore, calibration to 45-day periods may not capture the full range of processes that lead to hydrologic response. I recommend demonstrating that these parameters are consistent with a longer range calibration period. If this is not easily feasible within project constraints, I alternatively recommend discussing the assumptions and possible limitations behind this methodology decision.**

We thank the referee for the question and recommendations. Preliminary work was undertaken to arrive at the calibration settings documented in the manuscript. As computational resources preclude us from demonstrating consistency with a long-range calibration period, we will include the following information in the revised manuscript to elaborate on the calibration choices and its implications.

"Calibration of WRF-Hydro is computationally intensive and involves choices that may be aligned to the purpose of the simulations. Here we study the influence of lateral flow on seasonal timescales and hence the main purpose of calibration is to obtain better streamflow outcomes on monthly to seasonal timescales, rather that improved simulations of daily scale streamflow events. The streamflow in the domain primarily occurs in the cool season (May to October), and model simulations using default parameter values exhibit biases during these high flow months (Fig. 2). However, preliminary results showed that event-based calibration to high flow days did not translate to improved monthly flows indicating that it is necessary to use a period at least of the order of a month, which includes both high and low flow days at the four gauges for calibration. As a 45-day period is computationally feasible, and yields reasonable outcomes at monthly to seasonal timescales, this length of time is chosen for calibration. The daily streamflow data is smoothed by aggregating to 3-day flows to dampen the effect of individual high flow days."

**Minor Comments:**

3. **Figure 1: Consider adding the channel network (even if it is only higher order channels) to help the reader visualize the hydrologic connectivity.**

We thank the referee for the suggestion to improve the figure. We will revise Figure 1 as shown below to include the channel network.

[Figure]

Figure 1: (a) The WRF-Hydro model domain in southeast Australia. (b) Features in the domain. Background shading indicates topography, and the surface water catchments (black outlines) that drain into the streamflow gauges (blue triangles) used for calibration are marked on the map. The basins outlined in red (Upper Basins, Ovens, and Murray Riverina) are used to analyse the influence of lateral flow in basins with varying topographic characteristics. The network of major rivers (blue lines) based on data from Geoscience Australia (https://pid.geoscience.gov.au/dataset/ga/42343) are shown in panel b.

4. **Section 2.2.1, Geographic Data: I find it surprising that the TERN dataset produced worse streamflow compared to the default soil dataset. Is this something that could eventually be improved with calibration?**

We thank the referee for the question. We agree that it is surprising that the TERN dataset produced worse streamflow, and we had briefly discussed this in the original manuscript (lines 135-145). Based on our results,

rather than improving through calibration, we speculate that that it may be possible to obtained improved streamflow outcomes by utilising the TERN soil data in conjunction with regional soil parameter measurements.

We will improve upon lines 135-145 from the original manuscript as below.

"The better streamflow simulations obtained using the default soil dataset rather than the TERN dataset may be surprising, as the TERN data, which utilises regional observations (Teng et al., 2018), likely provides more accurate soil information over Australia. This is possibly because the modelled influence of soils on surface water partitioning in each land column relies on soil parameters in addition to soil type. The land model uses parameters (such as moisture at saturation, field capacity, wilting point, saturated hydraulic conductivity) for each soil type, the default values for which are defined based on scarcely available field observations in various regions (Kishné et al., 2017). Our results suggest that regional measurements that can be used to refine the default parameters values in conjunction with the regional soil datasets, such as the TERN dataset, may be necessary to obtain improved simulations."

5.  **Section 3.1.2, lines 215-220: The negative ET bias might reflect the limits of the 2m LSM. I recommend connecting back to this point in the discussion.**

We thank the referee for the suggestion to include the point about soil depth in the discussion. We will include this point in the paragraph detailing avenues of future work.

"…, more accurate representation of processes including baseflow and the feedback from the channels to the soil columns may be relevant in arid and semi-arid regions, such as southeast Australia. These processes are not represented in our simulations. Our calibration results suggest that spatially varying baseflow may improve lateral flow representation in our domain. Incorporating soil layers deeper than the 2-m depth modelled in standard Noah-MP have reduced surface flux biases in some cases in other semi-arid locations (Gochis et al., 2010; Barlage et al., 2015) and may be explored in future work. Incorporating channel seepages have improved streamflow simulations in semi-arid Arizona (Lahmers et al., 2021), but the feedback of this process to the soil column has not been modelled yet."

---

## Author Response (AR2)

We thank the editor and referees for the encouraging comments and suggestions to improve the manuscript.

We undertook minor revisions to incorporate the comments raised by referee 2. We have given point-by-point responses to the referees' comments and made necessary changes to the manuscript in response to suggestions. We believe that the revised manuscript fully satisfies the referees' concerns.

Referee comments are shown in bold. Author responses are shown in plain text.

**Referee 1**

**The authors have clearly addressed all of my concerns. I especially appreciate the changes that were made to the figures to make them more readable, and the clarification of the methods. Lateral water transfers (both surface and subsurface) are clearly very important in this complex terrain system on representation of ET. Great work!**

We thank Dr. Aaron Alexander for their thorough review and encouraging comments on the manuscript.

**Referee 2**

We thank Dr. Zhao Yang for their thorough review and suggestions to improve the manuscript.

**Specific Comments:**

**1. The authors should more clearly articulate the significance of lateral flow beyond the references already cited. While the current discussion is helpful, it would benefit from elaboration on how lateral flow becomes increasingly important at hyper-resolution scales. A critical assumption in many land surface models (LSMs) is that surface and subsurface runoff are removed from the system at the end of each time step. When lateral flow is explicitly represented, however, this water remains within the domain and continues to interact with the hydrological cycle. This distinction is crucial and should be emphasized in the introduction, as it fundamentally affects water and energy flux partitioning in LSMs (Yang et al., 2021).**

We thank the referee for the suggestion. We have included this point in the introduction of the revised manuscript.

Page 3, Lines 93-95:

"In standard LSM simulations, surface and subsurface runoff are removed from the system at the end of each model time step. In LSM simulations with lateral flow representation, runoff remains in the system and continues to affect other water cycle components such as ET and soil moisture."

**2. Paragraph starting 53, This paragraph makes a valuable attempt to summarize different representations of lateral flow, but the discussion is somewhat limited. Key contributions are missing, for instance, the work by Peter Hazenberg (Hazenberg et al., 2015a, 2015b), the Advanced Terrestrial Simulator (ATS), HydroGeoSphere, among others. The authors are encouraged to expand this section and provide a more comprehensive overview of existing modeling frameworks.**

We thank the referee for the suggestion.

Herzenberg et al. (2015a) and Hazenberg et al. (2015b) details a hillslope-based formulation with specified connectivity for land surface modelling, and we have now cited this work appropriately in the paragraph (page 2, lines 58-59).

Further, the referee refers to integrated surface subsurface hydrological models that incorporate lateral flow. There formulations have not been used in LSMs likely because of the challenges in coupling them. We believe that an introduction focussed on LSMs works better for this paper. We have now made it clear our summary in the paragraph starting on line 53 pertains only to lateral flow in LSMs.

Page 3, lines 70-74:

"It is worth noting that this summary pertains to modelling systems that includes lateral flow processes in LSMs used for Earth system modelling. Other models of lateral flow have been developed in the domain of integrated surface subsurface hydrological modelling to understand watershed system function (Bhanja et al., 2023; Brunner and Simmons, 2012). But these formulations have not been used in LSMs, likely due to the challenges in coupling them with LSMs."

**3. I believe the manuscript inaccurately states that subsurface flow is simulated on the LSM grid in WRF-Hydro. As far as I know, both surface and subsurface routing are handled on the routing grid. Please double-check this and revise accordingly.**

We thank the referee for noting this error. The referee is correct that both surface and subsurface routing are handled on the routing grid, and we have revised this.

Pages 4-5, lines 116-117:

"WRF-Hydro has the capability to simulate overland, shallow subsurface, and channel flows on the fine resolution routing grid."

**4. Line 289-295, The description of the "water balance" is somewhat misleading. What the authors present appears to be a partial water budget rather than a rigorous water balance analysis. A complete water balance requires quantification of all relevant terms (e.g., precipitation, ET, surface runoff, subsurface runoff, streamflow) and verification that the total inputs and outputs are conserved. Given that baseflow is disabled in the simulation, it is essential to demonstrate whether the budget closes with the available terms. The claim that ET and runoff exceed precipitation and that the difference can be attributed to soil moisture is problematic unless explicitly supported by budget closure analysis. Please provide evidence to substantiate this interpretation or revise the claim.**

We thank the referee for raising this intricate point. The water cycle terms we present are based on estimates that close the water balance. However, the referee is correct that the manuscript did not contain enough information to substantiate this. In the revised manuscript, we include more detail about the calculation of the water cycle terms and add Appendix Table A2 to support Table 3 in the manuscript by demonstrating water balance closure as detailed below.

In the Methods (page 9, lines 220-231):

"**2.4. Water balance in the simulations**
The simulated water cycle components are used to understand the influence of lateral flow on surface water partitioning. In control simulations using Noah-MP LSM without lateral flow, incoming precipitation is partitioned into ET, surface runoff (variable name: *sfcrnoff*), underground runoff (variable name: *udgrnoff*), and changes in soil moisture in the four layers (0-10 cm, 10-40 cm, 40 – 100 cm, 100 – 200 cm) of the soil column. The volumetric soil moisture in each layer converted to water depths are used to estimate the total soil moisture change for water balance calculations. The total runoff

is estimated in two ways (a) as the sum of the surface and underground runoff components, and (b) as the residual of precipitation after ET and soil moisture changes. We use these components to demonstrate the closure of the water balance in the control simulations. In simulations including lateral flow, the total runoff consists of overland flow, channel flow and underground runoff components simulated on the fine resolution routing grid. The runoff terms on the routing grid are not written to output files to reduce computational expense. Hence, we estimate the total runoff in the lateral flow runs as the residual of precipitation after ET and soil moisture changes, closing the water balance in a manner consistent with the control simulations."

**Table A2. Domain average water cycle terms accumulated over a 2-year period from 2015-12 to 2017-11 in the simulations. Negative soil moisture changes indicate a loss of moisture from the 2-m soil column over the 2-year period.**

| Variable | Simulation | | | | | | | | |
|---|---|---|---|---|---|---|---|---|---|
| | CTL1 | LAT1-100 | LAT1-250 | CTL4 | LAT4-100 | LAT4-250 | CTL10 | LAT1-100 | LAT1-250 |
| Precipitation, $P$ (mm) | 1504.4 | 1504.4 | 1504.4 | 1504.7 | 1504.7 | 1504.7 | 1503.1 | 1503.1 | 1503.1 |
| $ET$ (mm) | 1264.3 | 1307.7 | 1289.1 | 1262.9 | 1317.2 | 1308.1 | 1260.8 | 1309.6 | 1304.8 |
| Soil moisture change, $SM$ (mm) | -17.9 | -12.2 | -13.5 | -17.9 | -14.2 | -14.6 | -17.9 | -15.0 | -15.7 |
| Surface runoff, $sfcrnoff$ (mm) | 46.2 | | | 46.5 | | | 47.5 | | |
| Underground runoff, $udgrnoff$ (mm) | 211.8 | | | 213.2 | | | 212.7 | | |
| (a) Total runoff ($sfcrnoff + udgrnoff$), Ro (mm) | 258.0 | | | 259.7 | | | 260.2 | | |
| (b) Total runoff ($P - ET - SM$), $Ro$ (mm) | 258.0 | 208.9 | 228.8 | 259.7 | 201.7 | 211.2 | 260.2 | 208.5 | 214.0 |
| Water balance closure error, $P - ET - SM - sfcrnoff - udgrnoff$ (mm) | 0.0 | | | 0.0 | | | 0.0 | | |
| Runoff ratio ($Ro/P$) | 0.17 | 0.14 | 0.15 | 0.17 | 0.13 | 0.14 | 0.17 | 0.14 | 0.14 |

In the Results ():

"Appendix Table A2 lists the domain average precipitation, ET, runoff components, and the change in soil moisture from the nine simulations. In the control runs, total runoff estimated as (a) the sum of the surface runoff and underground runoff components matches (b) the residual of precipitation after ET and total soil moisture change, demonstrating the closure of the water balance. In simulations including lateral flow, total runoff is estimated as the residual of precipitation after ET and total soil moisture change."

We then go on to discuss the changes in partitioning due to including of lateral flow (which was part of the original manuscript).

Hazenberg, P., Y. Fang, P. Broxton, D. Gochis, G.-Y. Niu, J.D. Pelletier, P.A. Troch and X. Zeng, 2015a: A hybrid-3D hillslope hydrological model for use in Earth system models, Water Resour. Res., 51, doi: 10.1002/2014WR016842.

**Hazenberg, P., P. Broxton, D. Gochis, G.-Y. Niu, J.D. Pelletier, P.A. Troch, and X. Zeng, 2015b: Testing the hybrid-3D hillslope hydrological model in a controlled environment, Water Resour. Res., 52, 1089–1107, doi: 10.1002/2015WR018106.**

**Yang, Z., Huang, M., Berg, L. K., Qian, Y., Gustafson, W. I., Fang, Y., Liu, Y., Fast, J. D., Sakaguchi, K., and Tai, S.-L.: Impact of Lateral Flow on Surface Water and Energy Budgets Over the Southern Great Plains—A Modeling Study, Journal of Geophysical Research: Atmospheres, 126, e2020JD033659, https://doi.org/10.1029/2020JD033659, 2021.**